# PyHealth 2.0: A Comprehensive Open-Source Toolkit for Accessible and Reproducible Clinical Deep Learning

**John Wu** [1 2 3]  **Yongda Fan** [1 2]  **Zhenbang Wu** [1]  **Paul Landes** [4]  **Eric Schrock** [1 2]  **Sayeed Sajjad Razin** [2 5]
**Arjun Chatterjee** [1 2]  **Naveen Baskaran** [1 2]  **Joshua Steier** [2]  **Andrea Fitzpatrick** [1 2]  **Bilal Arif** [1 2]  **Rian Atri** [2]
**Jathurshan Pradeepkumar** [1 2]  **Siddhartha Laghuvarapu** [1 2]  **Junyi Gao** [6 7]  **Adam R. Cross** [4]  **Jimeng Sun** [1 3]

## Abstract

Difficulty replicating baselines, high computational costs, and required domain expertise create persistent barriers to clinical AI research. To address these challenges, we introduce PyHealth 2.0, an enhanced clinical deep learning toolkit that enables predictive modeling in as few as 7 lines of code. PyHealth 2.0 offers three key contributions: (1) a comprehensive toolkit addressing reproducibility and compatibility challenges by unifying 15+ datasets, 20+ clinical tasks, 25+ models, 5+ interpretability methods, and 5+ uncertainty quantification methods within a single framework that supports diverse clinical data modalities—signals, text, imaging, and electronic health records—with translation of 5+ medical coding standards; (2) accessibility-focused design accommodating multimodal data and diverse computational resources with up to 39× faster processing and 20× lower memory usage, enabling work from 16GB laptops to production systems; and (3) an active open-source community of 400+ members lowering domain expertise barriers through extensive documentation, reproducible research contributions, and collaborations with academic health systems and industry partners, including multi-language support via RHealth. PyHealth 2.0 establishes an open-source foundation and community advancing accessible, reproducible healthcare AI. Project details at https://pyhealth.dev/.

[1]University of Illinois Urbana-Champaign, Urbana, IL, USA [2]PyHealth Research Initiative [3]Keiji AI [4]University of Illinois College of Medicine, Chicago, IL, USA [5]Department of Biomedical Engineering, Bangladesh University of Engineering and Technology, Dhaka 1000, Bangladesh [6]The University of Edinburgh, Edinburgh, UK [7]Health Data Research UK, London, UK. Correspondence to: John Wu <johnwu3@illinois.edu>.

*Proceedings of the 43rd International Conference on Machine Learning*, Seoul, South Korea. PMLR 306, 2026. Copyright 2026 by the author(s).

## 1. Introduction

Regardless of the task, clinical deep learning models functionally follow a five-step pipeline: data processing, machine learning task definition, model initialization, training, and evaluation (Janiesch et al., 2021; Shinde & Shah, 2018; Wang et al., 2024). However, despite this standardized modeling process, reproducibility remains an increasingly difficult challenge due to both the rapid progress in AI developments and the lack of available, operable code (McDermott et al., 2019; 2021b; Beam et al., 2020). If any step is missing from this pipeline, reported results from clinical predictive models become practically irreproducible, making further development difficult. As such, it is crucial that all pipeline steps are available and reproducible.

**The Reproducibility Crisis.** Many clinical predictive models use similar data modalities—structured diagnosis and procedure codes, lab events—meaning their data processing often contains identical steps regardless of the task (Wang et al., 2024). Nonetheless, researchers consistently implement their own processing approaches with variations in implementation and random seeds, making reproducibility challenging (van de Water et al., 2024; Johnson et al., 2017). Furthermore, since each set of reported results is often associated with its own repository, directly auditing claims becomes tedious and unrewarding. When investigated, the majority of cohorts used in ML pipelines were found irreproducible (Johnson et al., 2017). Standardizing and centralizing these repeatable steps is key to improving pipeline reproducibility and transparency.

**Dependency and Compatibility Challenges.** Repositories often rely on dependencies that are defunct or incompatible with existing work environments (Hassan et al., 2024; Semmelrock et al., 2023). While Docker containers provide explicit research replication (Boettiger, 2015), they can be cumbersome, requiring additional dependencies and engineering practices beyond simple Python coding. Moreover, the goal of reproducibility extends beyond replication to exploring methods for developing better clinical AI models. A more reproducible future requires a unified framework of tested software compatible with standard working environ-

ments.

**Multimodal Complexity and Computational Barriers.** In recent years, the number of modalities considered within clinical AI systems has drastically increased, as electronic health records (EHR) are intrinsically multimodal (Acosta et al., 2022). From signals and images to lab events and codes, a unified framework must adapt to numerous differing modalities, as patient profiles are fundamentally multimodal. Large EHR datasets such as MIMIC-IV contain over 100 million lab events (Johnson et al., 2023), resulting in memory usage beyond conventional workstations. Our findings show memory requirements can balloon to several hundred gigabytes of RAM, making work on large clinical datasets highly infeasible on typical machines. Making healthcare AI more accessible requires reducing memory requirements for model training to fit within consumer-grade hardware.

**The Domain Knowledge Gap.** Finally, prerequisite domain knowledge creates barriers to auditing and understanding whether approaches are truly clinically relevant. As existing surveys evidence, a gap persists between the traditional machine learning and clinical communities (Nissar et al., 2023). The machine learning tasks developed should be crucial toward improving performance on problems highly relevant to real-world clinical needs. Achieving this requires not only technical expertise but also specific clinical experience to define available features and validate model performance. Bridging this cultural and technical gap between clinical experts and experienced AI researchers is crucial to ensure valuable human and computational resources are not wasted on the wrong problems.

**Our Contributions (Figure 1).** We introduce PyHealth 2.0, a deep learning toolkit which offers direct solutions to each of these problems through three key contributions: (1) **a comprehensive toolkit** spanning datasets, models, tasks, and evaluation methods that bridges the gap between technical and clinical domains through standardized implementations addressing reproducibility challenges; (2) **accessibility-focused design** that accommodates diverse computational resources and user backgrounds, achieving efficient memory usage and processing speed while supporting multiple programming languages; and (3) **an active open-source community** fostering reproducible healthcare AI research through collaborative development with over 50+ examples and tutorials that lower domain expertise barriers.

## 2. Related Works

PyHealth is not the only healthcare AI toolkit available for wider use. PyHealth 2.0's design draws inspiration from the broader healthcare AI reproducibility community, and

we acknowledge these contributions while highlighting PyHealth's distinct design philosophy and interoperability with existing frameworks. We provide an overview of the differences of PyHealth 2.0 with respect to many other health AI frameworks in Table 1.

**Other Health AI Frameworks.** Several specialized frameworks address different aspects of healthcare AI development. The **MEDS ecosystem** (McDermott et al., 2025) provides a suite of modular tools for developing and benchmarking models on longitudinal EHR data, operating as a network of interoperable packages linked by a common event stream data standard. Notable components include ACES for automatic cohort extraction (Xu et al., 2024) and MEDS-tab for automated baseline training of tabular models (Oufattole et al., 2024). **MONAI** (Cardoso et al., 2022) specializes in clinical imaging, offering tools for segmentation, classification, and generative modeling of medical images. **Zensols** (Landes et al., 2023) focuses on rapid reproduction of traditional clinical NLP models for text-based clinical problems.

PyHealth distinguishes itself through its flexibility and scope. By adopting a highly flexible event stream format (Arnrich et al., 2024), PyHealth maintains fundamental interoperability with these frameworks while offering unique advantages. Unlike MEDS's focus on longitudinal structured EHR data (McDermott et al., 2025), MONAI's specialization in imaging (Cardoso et al., 2022), or Zensols's emphasis on text (Landes et al., 2023), PyHealth's lack of datatype assumptions enables seamless integration of any combination of signals, clinical notes, lab events, structured medical codes, and images. Additionally, PyHealth requires only Python knowledge, eliminating the need to learn separate definition schemas such as those required by tools like ACES (Xu et al., 2024).

**Data Standards.** PyHealth makes no assumptions on data standards, and its flexible internal data structures support integration with multiple established clinical data standards. Common data models like OMOP (Reinecke et al., 2021), FHIR (Bender & Sartipi, 2013), i2b2 (Murphy et al., 2010), and PCORNet (Forrest et al., 2021) exist to standardize clinical vocabularies and longitudinal data storage across different institutions, making it easier to transfer models and perform data analysis across different institutions. PyHealth currently supports the OMOP standard directly while other data standards can be adapted without difficulty.

**Benchmarks.** Other software focus on benchmarking clinical models. YAIB (van de Water et al., 2024), EHR-PT (Mc-Dermott et al., 2021a), and a Multitask Clinical Benchmark (Harutyunyan et al., 2019) were constructed to standardize model performance benchmarking across popular clinical datasets. We note that PyHealth models are compatible with the majority of these benchmarks. One crucial differentiator

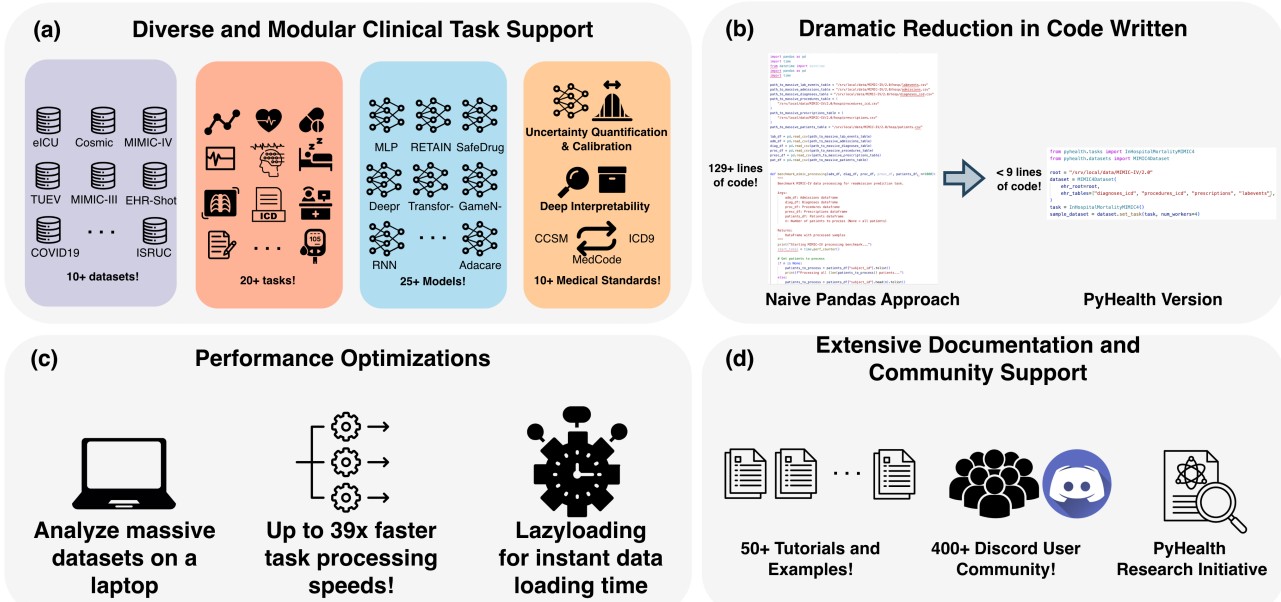

*Figure 1.* **PyHealth as a comprehensive healthcare AI development toolkit.** To tackle the reproducibility crisis in healthcare AI, (a) PyHealth has re-implemented over 10 different datasets for data loading, several dozen tasks, and over 25 different models with a variety of post-hoc model deployment features for better evaluating model performance. By re-implementing many of these components with standardized interfaces, (b) the amount of code required to generate samples ready for training on a specific ML task has been reduced from over 129 lines to fewer than 9. To enable users with limited computational resources, (c) PyHealth contains a variety of backend optimizations that enable data processing on modern laptops and support up to 39× faster processing speeds than alternatives (Figure 3). Finally, as a growing open-source community, (d) PyHealth 2.0 further embodies reproducible research principles through extensive examples and collaborative development.

here is that the PyHealth API enables easy modifications to existing defined benchmarks, making it flexible for a variety of clinical needs. As a key consequence, PyHealth can also be leveraged to quickly reproduce each of these benchmarks. Furthermore, PyHealth's evaluation goes beyond performance benchmarking, supporting model interpretability and uncertainty quantification, post-hoc analyses that enable a deeper look within a clinical model.

**What Makes PyHealth Different.** PyHealth's design philosophy centers on *comprehensiveness*: providing a unified framework that integrates diverse data modalities, model architectures, and analytical frameworks within a single cohesive system. This comprehensive approach offers several key advantages. First, researchers can work with multiple data types—structured EHR events, clinical notes, medical images, time series signals—without switching between specialized frameworks or managing complex dependency chains. Second, the unified API eliminates the overhead of learning multiple tools or reconciling different data formats across frameworks, allowing researchers to focus on model development rather than infrastructure. Third, by maintaining compatibility with established standards (OMOP (Reinecke et al., 2021), FHIR (Bender & Sartipi, 2013)) and interoperability with existing tools (MEDS (McDermott et al., 2025), MONAI (Cardoso et al., 2022)), PyHealth en-

ables integration into existing workflows while providing the convenience of a single, actively-maintained package.

As a comprehensive yet modular framework, PyHealth delivers integrated benefits spanning the complete modeling pipeline: optimized data processing, streamlined baseline reproduction, model interpretability tools, and uncertainty quantification methods. The open-source PyHealth community further strengthens this ecosystem through active peer-reviewed research that validates each contribution to the software package. Ultimately, PyHealth 2.0 provides a comprehensive suite of APIs that enables users to rapidly develop reproducible clinical models directly in their coding environments, facilitating adoption in both research and production settings.

## 3. PyHealth 2.0

PyHealth 2.0 currently consists of 9 major modules for comprehensive clinical model deployment, spanning from raw data processing to complex model evaluation, as shown in Figure 2. The PyHealth modules can be organized within two separate aspects of deep learning: model training and model evaluation. Each module has a set of base APIs that can be extended to support custom workflows.

**Model Training.** At its core, model training requires pro-

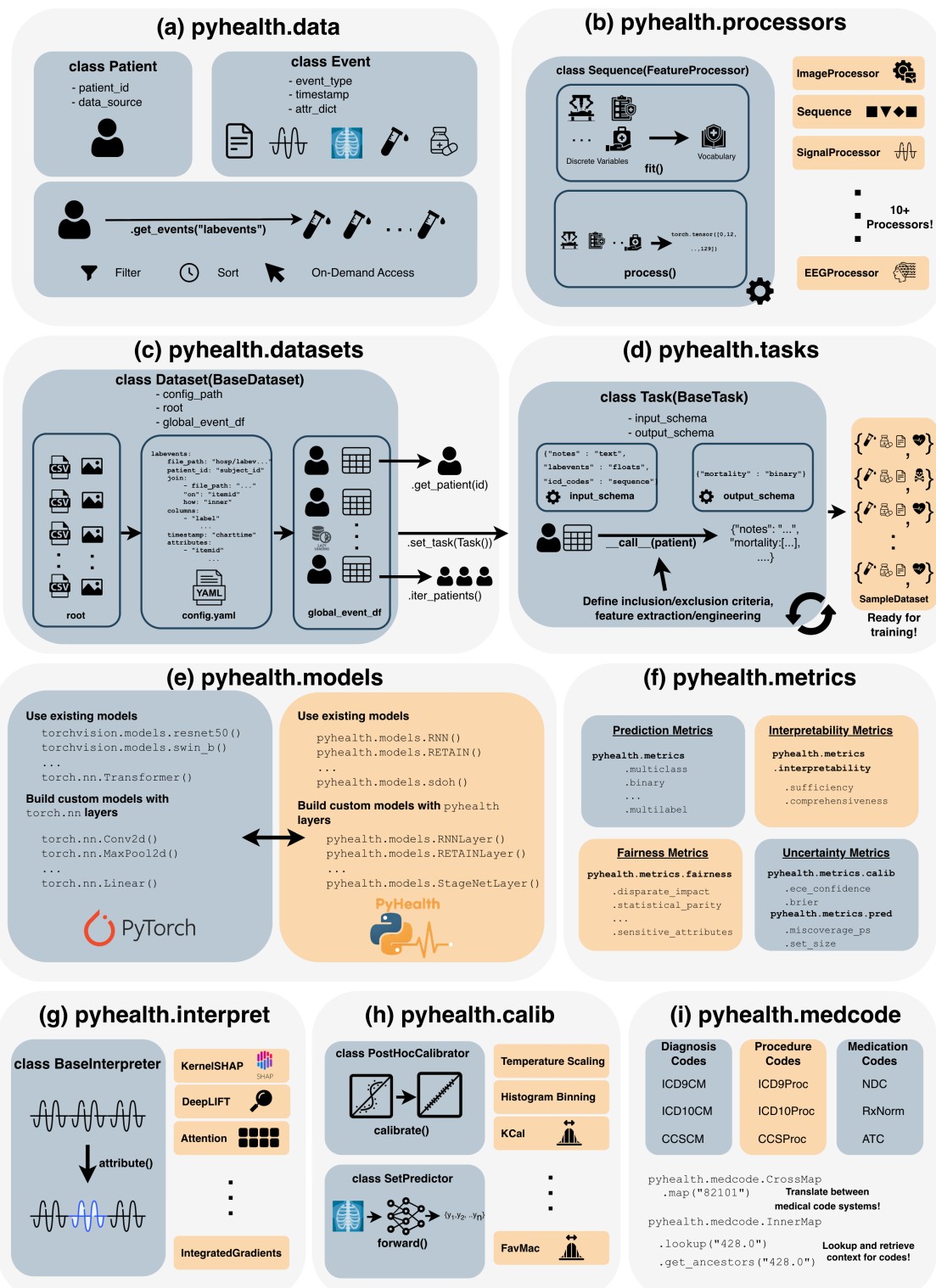

*Figure 2.* **PyHealth Overview.** PyHealth provides a comprehensive end-to-end pipeline for healthcare AI development spanning all aspects of clinical modeling. Starting from (a) standardized data structures (pyhealth.data) and (b) flexible data transformations (pyhealth.processors), we accommodate diverse clinical datasets through (c) memory-efficient lazy-loaded dataset loading (pyhealth.datasets) and (d) optimized task-specific ML processing for creating trainable formats (pyhealth.tasks). The framework supports (e) baseline model benchmarking (pyhealth.models) and (f) comprehensive evaluation metrics for fairness, uncertainty quantification, and interpretability—all critical for clinical deployment (pyhealth.metrics). For post-training utility, we provide (g) model interpretation tools (pyhealth.interpret), (h) uncertainty quantification via calibration and conformal prediction (pyhealth.calib), and (i) medical code translation and lookup across hospital coding standards (pyhealth.medcode).

*Table 1.* **Comparison of Healthcare AI Frameworks.** "Dynamically Scalable to Consumer Hardware" refers specifically to datasets that exceed the memory capacity of standard machines; frameworks without this support may still process large datasets given sufficient RAM on a workstation. Clinical imaging datasets often contain lightweight metadata files such that reading from disk occurs by design, meaning the memory footprint is usually quite small. In contrast, clinical EHR datasets contain massive tables that surpass memory constraints on even sizable workstations, which require further engineering practices. For examples of unified multimodal data support, interpretability, and uncertainty quantification, please see Appendix E.

| Feature | PyHealth 1.16 | MEDS | Zensols | MonAI | PyHealth 2.0 |
|---|---|---|---|---|---|
| Reproducibility Focused | ✓ | ✓ | ✓ | ✓ | ✓ |
| Biosignals | ✓ | ✗ | ✗ | ✗ | ✓ |
| EHR | ✓ | ✓ | ✗ | ✗ | ✓ |
| Clinical Text | ✗ | ✗ | ✓ | ✓ | ✓ |
| Clinical Imaging | ✗ | ✗ | ✗ | ✓ | ✓ |
| Unified Multimodal Data Support | ✗ | ✗ | ✗ | ✗ | ✓ |
| Large Selection of Models | ✓ | ✗ | ✗ | ✓ | ✓ |
| Interpretability | ✗ | ✗ | ✗ | ✓ | ✓ |
| Uncertainty Quantification | ✓ | ✗ | ✗ | ✗ | ✓ |
| Programming API Only | ✓ | ✗ | ✗ | ✓ | ✓ |
| End to End Pipelines | ✓ | ✗ | ✗ | ✓ | ✓ |
| Dynamically Scalable to Consumer Hardware | ✗ | ✓ | ✗ | ✗[*] | ✓ |

cessing raw data into a trainable format (i.e torch tensors). To do so, PyHealth supports the following modules to take a raw dataset and perform training in 7 lines of code as shown in Figure 1.

**pyhealth.data** provides a flexible 2-layer hierarchical data structure. Like the MEDs schema (Arnrich et al., 2024), everything is organized around patients, where each Patient contains a set of Event objects. Unlike MEDs, PyHealth makes no assumptions about dates, data types, or formats, enabling support for diverse clinical datasets including signals, images, structured EHR, and other modalities. For datasets without patient_id, each sample or row is treated as a Patient with a single event.

**pyhealth.processors** enables rapid transformation between data types through normalization, tokenization, and other clinical model requirements. Most processors support direct translation of continuous or discrete variables into torch tensors for training.

**pyhealth.datasets** introduces an optimized lazy-loading solution in PyHealth 2.0 for both dataset exploration and task construction. By loading each patient into memory only when needed, entire dataset tables load nearly instantaneously while ML task processing dynamically adapts to different memory constraints. As task processing can be expensive across millions of events, our dataset class supports parallel task processing, enabling massive speedups as shown in Figure 3.

**pyhealth.tasks** provides a simple, readable interface for defining diverse machine learning tasks through three components: (1) an input schema specifying expected inputs and pyhealth.processors for sample transformation, (2) an output schema defining the objective (generation, classification, etc.), and (3) a call function describing the sample construction process (feature inclusion, engineering, transformations, etc.). This design ensures that practitioners can immediately understand the inputs, outputs, and feature inclusion logic of any clinical ML task simply by reading its implementation. Once defined, the set_task() function performs this task transformation in parallel, generating a SampleDataset object that inherits from a PyTorch Lightning streaming dataset class to ensure memory-efficient data iteration (Chaton & AI, 2023).

**pyhealth.models** are custom PyTorch (Paszke et al., 2019) models, making them interchangeable with any PyTorch model while providing the modularity needed to transition between frameworks. This design enables direct compatibility with distributed training frameworks such as PyTorch Lightning (Falcon, 2019).

**Model Evaluation.** PyHealth offers a variety of ways of evaluating a model from interpretability, predictive performance, uncertainty quantification, to even translating medical codes to better understand a model's inputs. Below, we discuss each of the modules that enable comprehensive evaluation of a deep clinical model.

**pyhealth.metrics** offers comprehensive model evaluation beyond standard classification metrics for binary, multiclass, and multilabel settings. The module includes fairness metrics, interpretability metrics, and uncertainty quantification metrics to assess how fair, interpretable, and well-calibrated models are—critical considerations for clinical deployment.

**pyhealth.interpret** provides qualitative feature attribution analysis to complement quantitative evaluation. Interpreting deep learning models typically requires direct layer access (attention weights, convolution feature maps, intermediate gradients), specific input formatting, or navigation of incompatible dependencies across libraries like SHAP (Lundberg & Lee, 2017) and Captum (Kokhlikyan et al., 2020). To address this, we directly implement popular interpretability approaches—including Attention-Grad (Chefer et al., 2021), GIM (Edin et al., 2025), DeepLift (Shrikumar et al., 2017), and SHAP (Lundberg & Lee, 2017)—for seamless integration with clinical models.

**pyhealth.calib** addresses uncertainty quantification, which is critical for deployment in high-risk clinical settings. The module supports multiple techniques including model calibration (Wang, 2023; Guo et al., 2017) and conformal prediction (Angelopoulos et al., 2023).

**pyhealth.medcode** handles medical coding standard translations. Clinical data uses various standards, most commonly diagnosis codes like International Classification of Diseases (ICD) (Cartwright, 2013) and Clinical Classification Software (CCS) (Wei et al., 2017), or drug codes like ATC (Miller & Britt, 1995), RXNorm (Nelson et al., 2011), and NDC (Simonaitis & McDonald, 2009). PyHealth supports direct translation between these ontologies plus definition and ancestor lookups for contextualizing code inputs and predictions.

We further describe other implementation details in Appendix section C and how users can extend PyHealth for their own use cases in Appendix sections G, H, I.

## 4. Results

**PyHealth 2.0 offers substantial improvements over PyHealth 1.16 in functionality and accessibility.** To drive adoption, a software repository must provide compelling capabilities that justify its use. As discussed in Table 1, PyHealth 2.0 introduces several key advantages over existing healthcare AI frameworks. Unlike MEDS (McDermott et al., 2025), MonAI (Cardoso et al., 2022), and PyHealth 1.16 (Yang et al., 2023a), it supports true multimodal data integration—combining images, biosignals, structured codes, and clinical notes within a single dataloader. This enables researchers to explore richer feature combinations in their clinical pipelines.

Beyond data handling, PyHealth 2.0 provides an expanding model library and comprehensive post-hoc deployment tools including interpretability and uncertainty quantification. The updated toolkit scales dynamically across different hardware configurations. Most significantly, PyHealth 2.0 addresses the memory management issues identified by Steinberg et al. (2024) in PyHealth 1.16, enabling users to train clinical predictive models on consumer-grade hardware where memory is limited.

**Performance Benchmarking Setup.** To demonstrate the performance benefits of PyHealth 2.0, we benchmark on MIMIC-IV version 2.2 (Johnson et al., 2023) using an AMD EPYC 7513 32-core processor workstation with 1TB of RAM. This dataset contains 315,460 patients, 454,324 admissions, 5,006,884 diagnosis codes, 704,124 procedure codes, 124,342,638 lab events, and 669,186 drug codes. We evaluate three tasks: mortality prediction (using lab events and ICD codes), drug recommendation, and length-of-stay prediction. Each task requires joining multiple tables and iteratively extracting patients who have all necessary events. Of these, mortality prediction is the most expensive in both time and memory due to its use of lab events.

**PyHealth 2.0 scales efficiently from laptops to large compute clusters.** Addressing the reproducibility crisis requires lowering barriers to entry for training clinical AI models. PyHealth 2.0 achieves this through substantial performance improvements: task processing scales more efficiently across multiple workers than PyHealth 1.16, using significantly less memory while better utilizing available CPU cores for data transformation. Critically, our benchmark demonstrates end-to-end efficiency—the system not only extracts patient cohorts but directly translates and caches data into trainable tensor formats, making subsequent task reuse instantaneous after initial processing. In Figure 3, PyHealth 2.0's memory usage remains relatively constant across worker counts, and is consistently faster at ML task processing than the naive Pandas approach and its previous PyHealth 1.16 version across the majority of worker counts across all three tasks. We observe that the more modular MEDS counterpart, which includes the use of two frameworks MEDS_ETL and MEDS_reader (Steinberg et al., 2024), while substantially more memory-efficient than PyHealth 1.16, trades this memory efficiency for less multi-core performance.

**Consumer hardware outperforms server infrastructure with PyHealth 2.0.** A striking finding from our laptop task processing benchmarks is that PyHealth 2.0 running on a 16 GB MacBook Pro M2 Pro is *faster* on the labevents mortality task than our 32-core server workstation. This arises from three hardware factors: (1) the workstation accesses data over a NAS, introducing network I/O bottlenecks, whereas laptops read from local NVMe SSDs; (2) laptops offer higher memory bandwidth (unified LPDDR5/DDR5 vs.

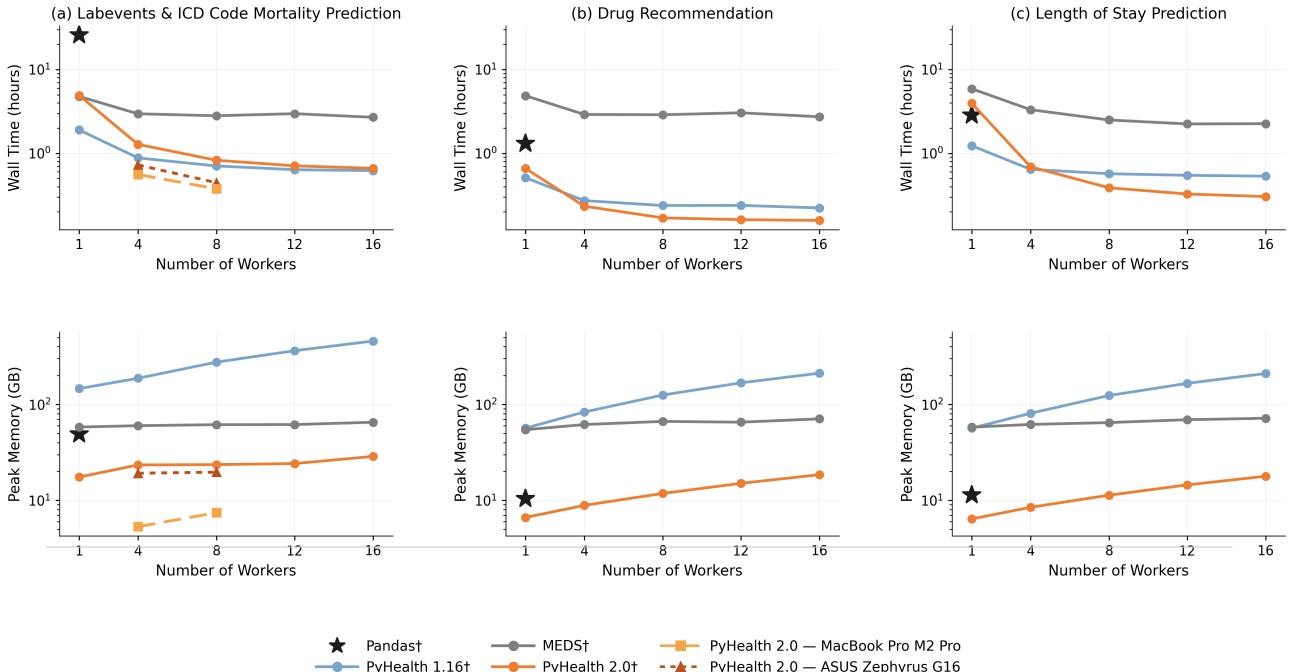

*Figure 3.* **PyHealth 2.0 delivers efficient scaling across machines.** We benchmark PyHealth 2.0 against PyHealth 1.16, MEDS, and a naive Pandas baseline — each processing raw MIMIC-IV (Johnson et al., 2023) `.csv` files through to trainable tensor formats — across three tasks: (a) labevents-based mortality prediction, (b) drug recommendation, and (c) length of stay prediction. All baselines and the Large Workstation (WS, NAS, DDR4) runs use a server-class machine; panel (a) additionally reports PyHealth 2.0 results on a MacBook Pro M2 Pro (16 GB) and an ASUS Zephyrus G16 (32 GB DDR5). PyHealth 2.0 achieves up to 39× speedup over naive Pandas and remains comparable to or faster than PyHealth 1.16 on wall time. Crucially, PyHealth 2.0 running on a consumer MacBook Pro M2 Pro consumes over 35× less peak memory than PyHealth 1.16 on a server-class workstation, a platform that previously required dedicated server infrastructure now fits comfortably within a 16 GB laptop memory budget. This is enabled by multi-worker sharding via Dask (Rocklin et al., 2015) and Polars (Nahrstedt et al., 2024), which keep memory consumption nearly constant as worker count scales. Exact numbers are provided in Appendix B. † Pandas, PyHealth 1.16, and MEDS are benchmarked on the WS only.

DDR4); and (3) PyHealth 2.0's sequential streaming workload is bottlenecked by single-core throughput, where modern consumer hardware excels. At 8 workers, the MacBook Pro completes the full labevents task in 1,357 s—faster than PyHealth 1.16 at 16 workers (2,235 s) on the server—while consuming a fraction of the memory. Researchers without dedicated server infrastructure can therefore achieve competitive or superior throughput on a standard laptop, substantially broadening access to clinical AI development.

**PyHealth 2.0 offers streamlined patient data exploration.** Exploring patient data in large disaggregated EHR datasets requires aggregating event data from multiple sources. This involves joining multiple tables, querying a specific patient, and loading each relevant event into a single data structure. Our updated PyHealth API has further streamlined patient data exploration, reducing the required code to aggregate all patient events for a given patient in MIMIC-IV (Johnson et al., 2023) from 14 lines in PyHealth 1.16 (Yang et al., 2023a) to only 10 in our 2.0 release, as shown in Table 2 and Figure 4 (a). With a single .get_events() call, users can explore all patient events from the specified tables lazy-loaded by the dataset object.

**Defining a new reproducible ML task is straightforward with PyHealth.** A key goal in improving healthcare AI accessibility is reducing the complexity of defining tasks for downstream training. In Table 2, the number of lines of code to use an already-implemented task remains constant regardless of task type. To define a new task, only a single function call is required to leverage all optimized backend processing, as shown in Figure 4 (b). By standardizing how each task is called within the rest of the pipeline (Figure 4 (c)), users can rapidly prototype and experiment with different features for modeling purposes. In Appendix E, we demonstrate how users can quickly define a mortality prediction task that incorporates all modalities within MIMIC-IV (Johnson et al., 2023), including clinical notes, X-rays, lab events, and structured EHR codes. In Appendix F, we show benchmark results across various healthcare ML tasks, uncertainty quantification methods, and interpretability approaches.

*Table 2.* **PyHealth 2.0 standardizes clinical pipeline deployment with fewer lines of code.** Compared to Pandas, PyHealth 1.16, and the MEDS ecosystem (MEDS_etl (McDermott et al., 2025) and MEDS_Reader (Steinberg et al., 2024)), PyHealth 2.0 achieves the fewest lines for patient data exploration and a uniform 7-line initialization across all tasks. PyHealth 1.16 yields slightly fewer lines on ML tasks due to its functional (non-OOP) style; by contrast, PyHealth 2.0's OOP design assigns each task variant a **unique identifier** (the primary source of additional lines), which is essential for reproducibility and extensibility as tasks and modalities scale. **Bold**: lowest count per column; ↓: improvement over Pandas and MEDS. *Pre-implemented task logic accounts for most ML code savings; custom tasks require only one additional function (Figure 4).

| Method | Patient Exploration | Mortality Prediction* | Length of Stay* | Drug Recommendation* |
|---|---|---|---|---|
| Pandas | 16 | 51 | 22 | 24 |
| PyHealth 1.16 | 14 | **27** | **14** | **16** |
| MEDS ETL + MEDS_Reader | 12 | 43 | 38 | 39 |
| PyHealth 2.0 | **10**↓ | 34↓ | 18↓ | 23↓ |

**(a) Patient Data Exploration in ~10 Lines**

```
from pyhealth.datasets import MIMIC4Dataset

if __name__ == "__main__":
    dataset = MIMIC4Dataset(
        ehr_root="your_directory_path,
        ehr_tables=["patients", "admissions", "diagnoses_icd",
        "procedures_icd", "labevents"],
    )
    patient = dataset.get_patient("10014729")
    events = patient.get_events() # We have our events!
```

**(c) Different Tasks, Same API**

```
from pyhealth.datasets import MIMIC4Dataset
base_dataset = MIMIC4Dataset(
    ehr_root="/srv/local/data/physionet.org/files/mimiciv/2.2/", # Lazyload dataset!
    ehr_tables=["patients", "admissions",
        "diagnoses_icd", "procedures_icd", "labevents", "prescriptions"],
)

from pyhealth.tasks import LengthOfStayPredictionMIMIC4
from pyhealth.tasks import DrugRecommendationMIMIC4          # Define your task!
from pyhealth.tasks import MortalityPredictionStageNetMIMIC4

mor_samples = base_dataset.set_task(MortalityPredictionStageNetMIMIC4())
los_samples = base_dataset.set_task(LengthOfStayPredictionMIMIC4())   # Datasets ready
dr_samples = base_dataset.set_task(DrugRecommendationMIMIC4())        # for training!
```

**(b) Defining a custom task in one call!**

```
class MortalityPredictionMIMIC4(BaseTask):   # Schemas define output
    input_schema: Dict[str, str] = {                    formats!
        "conditions": "sequence", "procedures": "sequence", "drugs": "sequence",
    }
    output_schema: Dict[str, str] = {"mortality": "binary"}

    def __call__(self, patient: Any) -> List[Dict[str, Any]]: # Define custom logic
        samples = []                                                      here!
        visits = patient.get_events(event_type="admissions")
        if len(visits) <= 1:
            return []
        for i in range(len(visits) - 1):  # Extracting specific
            visit = visits[i]                     event features!
            next_visit = visits[i + 1]
            mortality_label = int(next_visit.hospital_expire_flag)
            diagnoses = patient.get_events(
                event_type="diagnoses_icd", filters=[("hadm_id", "==", visit.hadm_id)]
            )
            conditions = [event.icd9_code for event in diagnoses]
            ... # Rest of logic code
            samples.append(
                {
                    "hadm_id": visit.hadm_id,
                    "patient_id": patient.patient_id,
                    "conditions": conditions,
                    "procedures": procedures_list,
                    "drugs": drugs,
                    "mortality": mortality_label,
                }
            )
        return samples
```

*Figure 4.* **Defining your own PyHealth tasks.** (a) Exploring patient data is key to defining your own custom task. (b) Once a task class is defined, PyHealth 2.0's optimized backend can handle the rest of the efficient parallel data processing. The key approach here is that for any new task, only a single class call has to be defined, following the (c) same pipeline for generating each task here.

## 5. Discussion

PyHealth 2.0 addresses fundamental barriers in clinical AI through unified multimodal capabilities, accessibility-focused design, and reproducible research infrastructure. These advances establish a foundation for tackling long-standing challenges in clinical AI deployment and reproducibility.

**Moving accessibility beyond Python-based communities.** Much of the bioinformatics community conducts research in R (Staples, 2023), limiting the ability to replicate and advance AI models on genomics (Akalin, 2020) and other modalities crucial to patient outcomes. To bridge this gap, the PyHealth community has introduced RHealth (Song et al., 2025), which brings PyHealth's core functionalities to R. As PyHealth develops, its design principles will extend to communities beyond Python-based machine learning.

**Future of Agentic Systems and PyHealth.** As coding agents rapidly improve (Yang et al., 2024; Wang et al., 2025b), PyHealth's unified API provides an ideal substrate for automated clinical AI pipelines. Its modality-agnostic design—making minimal assumptions about data types or architectures—enables agents to automate more of the clinical deployment workflow. While data privacy prevents exact replication across healthcare systems, well-documented code serves as a practical recipe for institutional adaptation. PyHealth thus functions as a centralized repository of recipes for agentic systems.

**PyHealth Roadmap.** PyHealth development remains actively ongoing with many planned features. While the number of models, datasets, and tasks continues to expand as the community grows, several features orthogonal to ML pipelines are under development. PyHealth 2.0 currently supports various interpretability and uncertainty quantifica-

tion methods as shown in Appendix Figure 6, though current implementations have several limitations.

First, many interpretability approaches assume access to specific PyHealth model attributes, such as embedding layers or hooks enabling gradient logging. These assumptions limit model compatibility with the interpretability module, currently restricting support to tabular time-series clinical predictive models and image-based models. Support for interpreting language models and other modalities is a major future priority as the field of mechanistic interpretability develops (Rai et al., 2024; Fiotto-Kaufman et al., 2024).

Second, many uncertainty quantification approaches such as conformal prediction (Angelopoulos et al., 2023; Papadopoulos et al., 2007) fail on clinical tasks because patient distribution shifts are common. While approaches like conformal prediction under covariate shift exist within the repository, their practical utility remains limited on various clinical tasks. As a key priority, PyHealth intends to explore personalized conformal prediction approaches to make the uncertainty quantification module more useful where distribution assumptions are unlikely to hold.

Finally, while PyHealth directly supports mixing and matching any modality as shown in Appendix Figure 5, its support for multimodal models capable of handling heterogeneous features remains highly limited. Though capable of extracting embeddings from HuggingFace (Jain, 2022) models and various provided unimodal PyTorch models, PyHealth does not readily provide recipes for off-the-shelf multimodal clinical predictions, especially under cases of missing modalities—highly prevalent in patient data (Wu et al., 2024). Addressing this limitation is crucial for improving software usability and adoption.

**Limitations.** Beyond the modality, interpretability, and uncertainty quantification gaps outlined in our roadmap, two further limitations warrant attention. First, our processing and efficiency benchmarks are sensitive to hardware and operating system: storage medium, memory bandwidth, and core count each meaningfully affect wall-clock time and peak memory, so reported gains may vary across deployments. Second, PyHealth 2.0 deliberately scopes itself to data processing and modeling and does not provide data governance or de-identification tooling; users remain responsible for regulatory compliance (e.g., HIPAA, GDPR) and the ethical handling of patient data. Component maturity is similarly uneven—data processing, task schemas, and evaluation are extensively tested, and EHR and EEG modeling are well-supported, while clinical imaging, text, and genomics currently provide only basic baselines—and we will continue to document this clearly as the framework matures.

**Concluding Statements.** We present PyHealth 2.0, a vastly improved comprehensive clinical deep learning toolkit. This software enables reproducible development of clinical pipelines in a fast and accessible manner. Our hope is that by contributing this repository to the wider community, we can incentivize sharing of reproducible models and, more importantly, operable code in the AI for healthcare community.

## Impact Statement

PyHealth aims to reduce the complexity of developing clinical AI models. In doing so, PyHealth has reduced computational requirements, improving the accessibility of building clinical AI models. Our goal is to democratize the ability to work in clinical AI.

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

## A. Implementation Details

All code is shared in the supplementary material, specifically with our Pandas and PyHealth 1.16 comparisons. We share our code through `https://anonymous.4open.science/r/PyHealth-CB8D/`, and recommend installing PyHealth 1.16 through `pip install pyhealth==1.16`. To reimplement our MEDS baseline, please install `meds_etl` and `meds_reader` through `pip install meds_etl meds_reader` respectively. For MEDS_etl, we use the default settings as denoted by their GitHub settings, which assumes 100 shards for caching. We vary the number of processors in conjunction with our framework to better understand how each framework scales across different compute levels.

## B. Explicit Benchmark Numbers for ML Task Processsing

We present exact performance numbers here for our three performance benchmarks from loading all of the raw data to final task processing in Tables 4, 5, and 3. We observe that PyHealth 2.0 exhibits the lowest peak memory usage while generally being comparable or even faster than PyHealth 1.16's approach despite PyHealth 1.16 loading everything in memory for parallel processing.

*Table 3.* Performance comparison for in-hospital mortality prediction on MIMIC-IV. Wall time in seconds, memory in GB. Best results per metric in bold. † PyHealth 2.0 laptop rows benchmarked at 4 and 8 workers only.

| Method | Number of Workers | | | | |
| --- | --- | --- | --- | --- | --- |
| | 1 | 4 | 8 | 12 | 16 |
| *Wall Time (seconds)* | | | | | |
| Pandas | 93,708 | ✗ | ✗ | ✗ | ✗ |
| PyHealth 1.16 | **6,841** | 3,174 | 2,543 | **2,295** | **2,235** |
| MEDS | 17,202 | 10,672 | 10,117 | 10,707 | 9,694 |
| PyHealth 2.0 (Large WS) | 17,671 | 4,605 | 2,975 | 2,551 | 2,385 |
| PyHealth 2.0 (MacBook Pro M2 Pro)† | — | **2,010** | **1,357** | — | — |
| PyHealth 2.0 (ASUS Zephyrus G16)† | — | 2,617 | 1,605 | — | — |
| *Peak Memory (GB)* | | | | | |
| Pandas | 49.23 | ✗ | ✗ | ✗ | ✗ |
| PyHealth 1.16 | 146.23 | 187.21 | 275.73 | 363.13 | 457.42 |
| MEDS | 58.07 | 60.00 | 61.48 | 61.62 | 65.19 |
| PyHealth 2.0 (Large WS) | 17.48 | 23.39 | 23.57 | **24.19** | **28.70** |
| PyHealth 2.0 (MacBook Pro M2 Pro)† | — | **5.31** | **7.43** | — | — |
| PyHealth 2.0 (ASUS Zephyrus G16)† | — | 19.15 | 19.67 | — | — |

*Table 4.* Performance comparison for drug recommendation on MIMIC-IV (excluding lab events). Wall time in seconds, memory in GB. Best results per metric in bold.

| Method | Number of Workers | | | | |
| --- | --- | --- | --- | --- | --- |
| | 1 | 4 | 8 | 12 | 16 |
| *Wall Time (seconds)* | | | | | |
| Pandas | 4,725 | ✗ | ✗ | ✗ | ✗ |
| PyHealth 1.16 | **1,841** | 981 | 857 | 860 | 802 |
| MEDS | 17,544 | 10,477 | 10,398 | 10,968 | 9,840 |
| PyHealth 2.0 | 2,375 | **841** | **610** | **581** | **569** |
| *Peak Memory (GB)* | | | | | |
| Pandas | 10.41 | ✗ | ✗ | ✗ | ✗ |
| PyHealth 1.16 | 56.62 | 83.24 | 125.22 | 167.84 | 211.62 |
| MEDS | 54.45 | 61.66 | 66.47 | 65.26 | 70.72 |
| PyHealth 2.0 | **6.63** | **8.86** | **11.80** | **15.04** | **18.50** |

## C. Dataloading and Caching Implementation Details

The dataloading pipeline consists of three cached stages: table-joining, task-transformation, processor-transformation.

Table-joining merges multiple CSVs into a single comprehensive event table. We use Dask (Rocklin et al., 2015) to join all

*Table 5.* Performance comparison for length of stay prediction on MIMIC-IV (excluding lab events). Wall time in seconds, memory in GB. Best results per metric in bold.

| Method | Number of Workers | | | | |
|---|---|---|---|---|---|
| | 1 | 4 | 8 | 12 | 16 |
| *Wall Time (seconds)* | | | | | |
| Pandas | 10,269 | ✗ | ✗ | ✗ | ✗ |
| PyHealth 1.16 | **4,434** | **2,316** | 2,054 | 1,965 | 1,922 |
| MEDS | 21,236 | 11,947 | 9,003 | 8,077 | 8,135 |
| PyHealth 2.0 | 14,228 | 2,484 | **1,396** | **1,176** | **1,096** |
| *Peak Memory (GB)* | | | | | |
| Pandas | 11.40 | ✗ | ✗ | ✗ | ✗ |
| PyHealth 1.16 | 56.60 | 80.89 | 123.76 | 165.79 | 209.93 |
| MEDS | 57.95 | 61.82 | 64.66 | 69.30 | 71.71 |
| PyHealth 2.0 | **6.40** | **8.46** | **11.31** | **14.49** | **17.83** |

tables due to its out-of-core joining and sorting capabilities, wrapped by Narwhals to ensure a consistent, Polars-like API for PyHealth. The output is sorted by `patient_id` and cached as Parquet files, allowing downstream processes to utilize Parquet min-max statistics for faster grouping.

Task-transformation groups events by `patient_id` to create samples based on specific PyHealth tasks. We use Polars to process batches of 128 consecutive patients; this approach optimizes speed by leveraging data locality and row-group statistics while keeping the memory usage low. To prevent thread starvation in multi-worker settings, the Polars thread pool count is calibrated per worker. Resulting samples are cached in LitData binary format (Chaton & AI, 2023).

Processor-transformation converts the generated samples into tensors. The output is cached in LitData binary format. During PyTorch training, we employ LitData's `StreamingDataset` (Chaton & AI, 2023) to enable efficient data loading, subsampling, shuffling, and splitting.

## D. Dataloading and Caching Performance Comparison

In principle, most datasets are cached after processing, eliminating the high computational costs of initial data loading from raw .csv files. Once processed, iteratively exploring patient data and constructing new ML tasks becomes relatively fast. Table 6 examines the time required to join all MIMIC-IV (Johnson et al., 2023) tables and format them into patient representations with all events. Note that this excludes machine learning task feature engineering and filtering of patients without specific events.

Remarkably, our naive Pandas solution proves remarkably fastest for table joins and caching, with relatively fast patient access times. Nonetheless, PyHealth 2.0 serves as the nearest competitive alternative. A key design change that leads to slower random patient access compared to its predecessor PyHealth 1.16 and related software MEDS_reader (Steinberg et al., 2024) is the elimination of patient index assumptions, removing fast-access maps of patient data. This design change accommodates imaging and signal datasets where patient IDs are not necessarily available. Compared to Pandas, PyHealth 2.0 stores much of the patient data on disk rather than directly in memory, resulting in slightly slower patient access times. Nevertheless, patient access times remain in the millisecond range, enabling fundamentally fast patient exploration. Future work could explore efficient patient data mapping strategies. This benchmark uses the same code structure shown in Table 2.

*Table 6.* **Dataloading, cache, and random patient access times.** *We note that part of our dataloading time with MEDS_reader (Steinberg et al., 2024) includes the conversion time from PyHealth 1.16, making it ultimately slower in this comparison to PyHealth 1.16. Nonetheless, patient access times are remarkably fast here in all cases, being measured in milliseconds. Please note that this patient random access time does not necessarily result in worse as all patients are loaded serially in training.

| | Pandas | PyHealth 1.16 | MEDS ETL + MEDS_Reader | PyHealth 1.16 + MEDS_Reader | PyHealth 2.0 |
|---|---|---|---|---|---|
| Load & Cache Time (s) | 740.97 | 3548.90 | 10117.04 | 3766.31* | 1093.38 |
| Patient Access Time (ms) | 168.23 | 0.01 | 0.01 | 0.02 | 208.85 |

# E. Example Use Cases

To demonstrate PyHealth 2.0's extensive feature set, we present qualitative examples showcasing major framework capabilities: (1) multimodal patient dataloading and (2) integrated interpretability and uncertainty quantification for model evaluation.

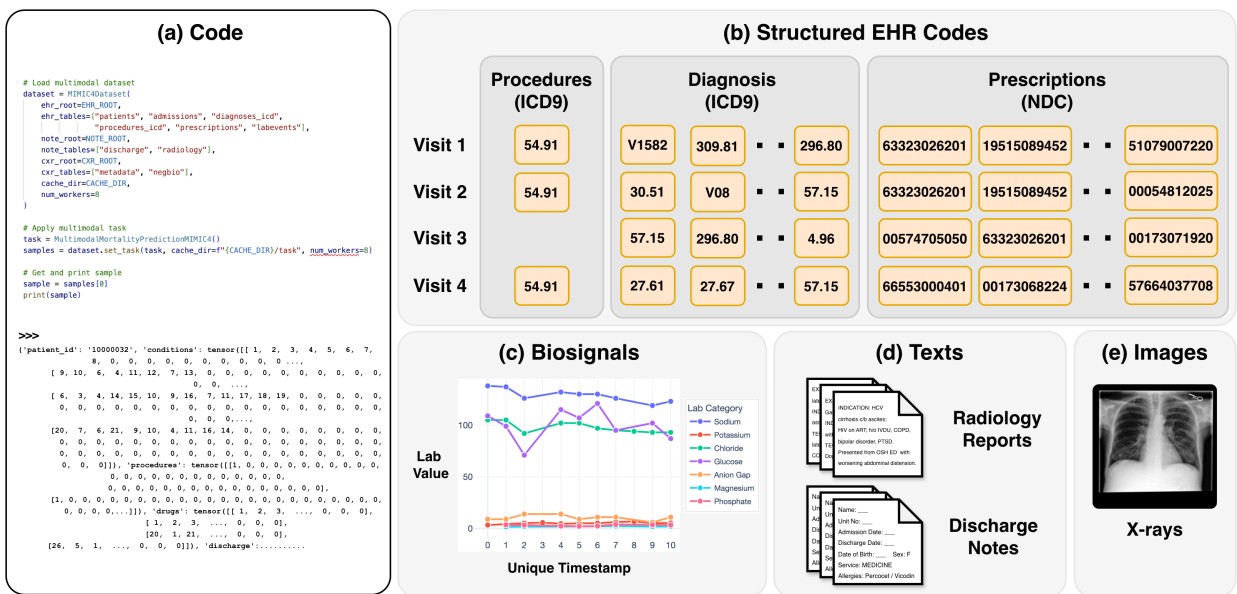

*Figure 5.* **PyHealth 2.0 directly supports loading multimodal data.** With effectively only 5 lines of code, PyHealth 2.0 now supports the ability to work with (b) structured EHR codes, (c) biosignals, (d) clinical notes, (e) and X-rays on MIMIC-IV data (Johnson et al., 2023).

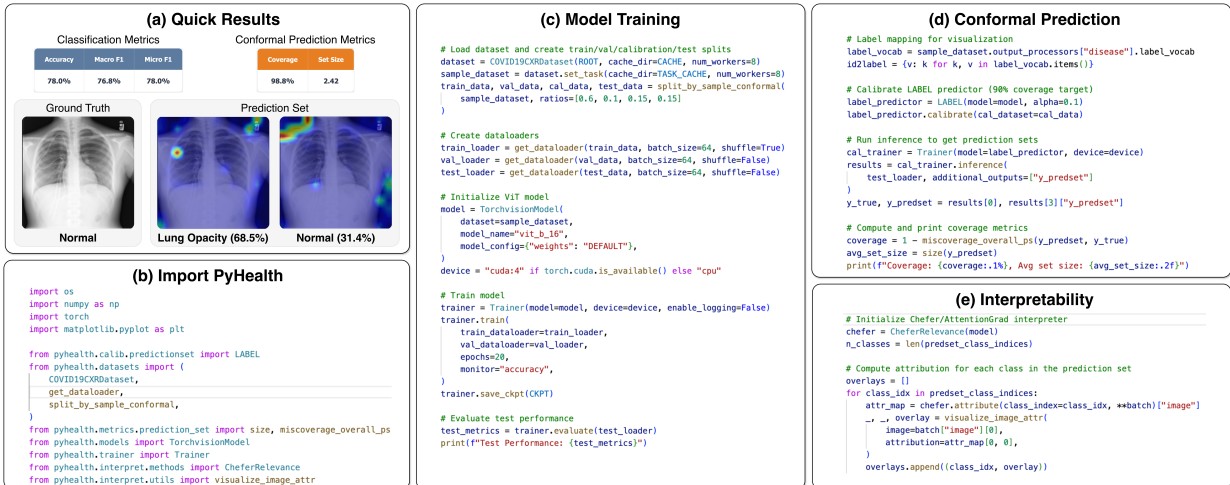

*Figure 6.* **PyHealth 2.0 supports various interpretability and uncertainty quantification deployment techniques.** To produce our results (a), we import PyHealth (b), train our model (c), finish conformal prediction (d), and run an interpretability visualization (e).

**Multimodal dataloading.** In Figure 5, we show how we can construct a dataset and explore an entire patient's extensive profile in MIMIC-IV. The MIMIC4Dataset.set_task() function is called on the MultiModalMortalityPrediction task object here, generating a sequence of multimodal events that quickly enables us to generate the visualizations in the rest of Figure 5.

**Interpretability and uncertainty quantification.** We fine-tune a Vision Transformer (ViT) on the COVID-19 Chest X-ray dataset (Rahman et al., 2021) to illustrate PyHealth's capacity for deeper model evaluation beyond standard predictive metrics. Clinical X-ray classification demands interpretable predictions to support diagnostic decision-making, while uncertainty quantification provides theoretical guarantees essential for building deployment trust. We configure conformal prediction with $\alpha = 0.01$ and evaluate empirical coverage through `pyhealth.metrics`. Subsequently, we apply

`pyhealth.interpretability` to analyze the fine-tuned ViT's decision-making robustness.

**Conformal interpretability results.** Our vanilla conformal prediction (Angelopoulos et al., 2023; Papadopoulos et al., 2007) achieves approximately 99% empirical coverage on the test set, consistent with the configured $\alpha = 0.01$ miscoverage rate. However, interpretability analysis using AttentionGrad (Chefer et al., 2021) reveals spurious feature dependence: the ViT incorrectly predicts "lung opacity" for a normal patient X-ray by primarily attending to annotation text in the image corner rather than anatomical features. While the conformal predictor correctly captures the ground truth as the second-most-likely class, the attention maps confirm reliance on spurious correlations here.

## F. Benchmark Results

To complement the data-processing benchmarks in the main paper, we provide downstream modeling results across every modality supported by PyHealth 2.0 (EHR, imaging, text, and biosignals), together with interpretability and uncertainty-quantification benchmarks. These results are intended as evidence that PyHealth 2.0's standardized task schemas and default configurations yield reasonable, reproducible baselines out of the box.

Unless otherwise noted, every model in this appendix is trained for **20 epochs** with **AdamW** (learning rate $1 \times 10^{-4}$) using the framework's default hyperparameters. We deliberately perform *no* hyperparameter tuning and *no* pretraining: the goal is to characterize default reproducible behavior rather than state-of-the-art performance.

### F.1. Datasets and Tasks

Benchmarks were collected on two compute servers. Table 7 summarizes the datasets and clinical tasks evaluated on each.

*Table 7.* Datasets and tasks benchmarked, grouped by compute server. Datasets: MIMIC-III (Johnson et al., 2016), MIMIC-IV (Johnson et al., 2023), eICU (Pollard et al., 2018), Covid19CXR (Rahman et al., 2021), TUAB (Obeid & Picone, 2016), and TUEV (Shah et al., 2018).

| Dataset | Task |
|---|---|
| *Server 1* | |
| MIMIC-III | Mortality Prediction |
| MIMIC-III | Readmission Prediction |
| Covid19CXR | Classification |
| MIMIC-IV | Mortality Prediction |
| MIMIC-IV | Length of Stay Prediction |
| MIMIC-IV | Drug Recommendation |
| MIMIC-IV | Readmission Prediction |
| *Server 3* | |
| eICU | Drug Recommendation |
| eICU | Length of Stay |
| eICU | Mortality Prediction |
| eICU | Readmission Prediction |
| Medical Transcriptions | Classification |
| TUH EEG (TUAB) | Abnormal EEG Detection |
| TUH EEG (TUEV) | EEG Event Classification |

## F.2. EHR Modeling

*Table 8.* Mortality prediction on MIMIC-III (Johnson et al., 2016).

| Model | AUROC | PRAUC | F1 | Accuracy | Loss |
|---|---|---|---|---|---|
| RNN | 0.5778 | 0.1378 | 0.0000 | 0.8967 | 0.3338 |
| StageNet | 0.5635 | 0.1323 | 0.0000 | 0.8967 | 0.3336 |
| Adacare | 0.5505 | 0.1520 | 0.0000 | 0.8967 | 0.3317 |
| Retain | 0.5493 | 0.1175 | 0.0000 | 0.8967 | 0.3419 |
| Transformer | 0.5415 | 0.1268 | 0.1152 | 0.8236 | 1.5007 |
| ConCare | 0.5043 | 0.1080 | 0.0000 | 0.8967 | 0.3381 |

*Table 9.* Readmission prediction on MIMIC-III (Johnson et al., 2016).

| Model | AUROC | PRAUC | F1 | Accuracy | Loss |
|---|---|---|---|---|---|
| ConCare | 0.5576 | 0.2100 | 0.0000 | 0.8278 | 0.4630 |
| RNN | 0.5549 | 0.1955 | 0.0000 | 0.8257 | 0.4633 |
| Adacare | 0.5500 | 0.2139 | 0.0000 | 0.8278 | 0.4584 |
| Retain | 0.5500 | 0.1967 | 0.0000 | 0.8236 | 0.4707 |
| TCN | 0.5423 | 0.2075 | 0.1370 | 0.8027 | 0.5157 |
| Transformer | 0.5296 | 0.1785 | 0.0221 | 0.8152 | 0.5173 |

*Table 10.* Mortality prediction on MIMIC-IV (Johnson et al., 2023).

| Model | AUROC | PRAUC |
|---|---|---|
| Deepr | 0.7547 | 0.0692 |
| RNN | 0.7481 | 0.0673 |
| StageNet | 0.7422 | 0.0791 |
| Retain | 0.7250 | 0.0588 |
| Transformer | 0.7109 | 0.0502 |
| ConCare | 0.6879 | 0.0540 |
| Adacare | 0.6653 | 0.0425 |
| EHRMamba | 0.6224 | 0.0456 |

*Table 11.* Length of stay prediction on MIMIC-IV (Johnson et al., 2023).

| Model | Accuracy | F1 Macro | Loss |
|---|---|---|---|
| RNN | 0.4567 | 0.4050 | 1.3752 |
| Retain | 0.4341 | 0.3792 | 1.4364 |
| Transformer | 0.4202 | 0.3529 | 1.5486 |
| Deepr | 0.4144 | 0.3511 | 1.4949 |
| EHRMamba | 0.3995 | 0.3349 | 1.5475 |
| Adacare | 0.3825 | 0.3142 | 1.6043 |

*Table 12.* Drug recommendation on MIMIC-IV (Johnson et al., 2023).

| Model | Jaccard | F1 Samples | Loss |
|---|---|---|---|
| EHRMamba | 0.4915 | 0.6503 | 0.0602 |
| Deepr | 0.4813 | 0.6408 | 0.0616 |
| Transformer | 0.4661 | 0.6266 | 0.0632 |
| Adacare | 0.4659 | 0.6268 | 0.0629 |
| RNN | 0.4301 | 0.5929 | 0.0677 |
| GameNet | 0.4278 | 0.5910 | 0.0690 |
| Retain | 0.4234 | 0.5873 | 0.0701 |

*Table 13.* Mortality prediction on eICU (Pollard et al., 2018).

| Model | AUROC | PRAUC | F1 | Accuracy | Loss |
|---|---|---|---|---|---|
| Retain | 0.6836 | 0.1807 | 0.0071 | 0.9183 | 0.2656 |
| Transformer | 0.6696 | 0.1847 | 0.0530 | 0.9159 | 0.2788 |
| RNN | 0.6693 | 0.2136 | 0.1859 | 0.9150 | 0.2850 |
| ConCare | 0.6306 | 0.1448 | 0.0000 | 0.9180 | 0.2773 |
| Adacare | 0.6248 | 0.1315 | 0.0000 | 0.9180 | 0.2781 |

*Table 14.* Length of stay prediction on eICU (Pollard et al., 2018).

| Model | Accuracy | F1 Macro | Loss |
|---|---|---|---|
| RNN | 0.3912 | 0.2420 | 1.5503 |
| Retain | 0.3796 | 0.1955 | 1.5876 |
| Transformer | 0.3696 | 0.1719 | 1.6659 |
| Adacare | 0.3547 | 0.1609 | 1.6569 |
| ConCare | 0.3491 | 0.1545 | 1.6655 |

*Table 15.* Readmission prediction on eICU (Pollard et al., 2018).

| Model | AUROC | PRAUC | F1 | Accuracy | Loss |
|---|---|---|---|---|---|
| Transformer | 0.8151 | 0.8570 | 0.7497 | 0.7353 | 0.5357 |
| RNN | 0.8071 | 0.8505 | 0.7523 | 0.7346 | 0.5475 |
| Retain | 0.8058 | 0.8498 | 0.7470 | 0.7319 | 0.5287 |
| Adacare | 0.7650 | 0.8140 | 0.7050 | 0.6866 | 0.5744 |
| ConCare | 0.7467 | 0.8014 | 0.7070 | 0.6756 | 0.6029 |

## F.3. Clinical Imaging

*Table 16.* Chest X-ray classification on Covid19CXR (Rahman et al., 2021). ROC-AUC is computed one-vs-rest (OvR).

| Model | Accuracy | F1 Macro | ROC-AUC (OvR) |
|---|---|---|---|
| CNN | 0.8649 | 0.8741 | 0.9750 |
| ResNet-18 | 0.9532 | 0.9601 | 0.9942 |
| ViT-B/32 | 0.8880 | 0.8926 | 0.9773 |

## F.4. Clinical Text

*Table 17.* Medical Transcriptions classification.

| Model | Accuracy | F1 Macro |
|---|---|---|
| Text Embedding | 0.3400 | 0.0664 |
| Text Embedding (BERT Base) | 0.3622 | 0.0470 |
| Text Embedding (BioBERT) | 0.3702 | 0.0718 |

## F.5. Biosignals (EEG)

*Table 18.* Abnormal EEG detection on the TUH Abnormal EEG Corpus (TUAB) (Obeid & Picone, 2016).

| Model | ROC-AUC | Accuracy | F1 Macro |
|---|---|---|---|
| BIOT | 0.8923 | 0.8099 | 0.7695 |
| ContraWR | 0.8334 | 0.7619 | 0.7278 |
| SPaRCNet | 0.8552 | 0.7677 | 0.6996 |
| TFM-Tokenizer | 0.8903 | 0.8113 | 0.7783 |

*Table 19.* Event classification on the TUH EEG Events Corpus (TUEV) (Shah et al., 2018).

| Model | Accuracy | F1 Macro |
|---|---|---|
| BIOT | 0.7162 | 0.4218 |
| ContraWR | 0.7411 | 0.5026 |
| SPaRCNet | 0.6909 | 0.3642 |
| TFM-Tokenizer | 0.7768 | 0.5451 |

## F.6. Non-EHR Data-Processing Benchmarks

Table 20 reports the data-processing cost (dataset loading, task processing, and peak memory) for the imaging, text, and biosignal datasets, complementing the EHR processing benchmarks in the main paper.

*Table 20.* Data-processing benchmarks for non-EHR modalities: Covid19CXR (Rahman et al., 2021), Medical Transcriptions, TUAB (Obeid & Picone, 2016), and TUEV (Shah et al., 2018). Wall time is the sum of dataset loading and task processing.

| Modality | Dataset | Loading (s) | Task Processing (s) | Peak Mem. (GB) |
|---|---|---|---|---|
| Image | Covid19CXR | 11.74 | 131.05 | 1.95 |
| Text | Medical Transcriptions | 16.73 | 31.73 | 2.24 |
| Signals (EEG) | TUAB | 0.60 | 6547.18 | 2.05 |
| Signals (EEG) | TUEV | 0.19 | 1210.60 | 2.78 |

## F.7. Interpretability

We additionally benchmark post-hoc interpretability methods on a Transformer model trained for mortality prediction, reported using the comprehensiveness and sufficiency faithfulness metrics. Best values per metric are shown in **bold** (higher comprehensiveness and lower sufficiency are better).

*Table 21.* Interpretability benchmarks for mortality prediction on MIMIC-IV (Johnson et al., 2023) with a Transformer model. Higher comprehensiveness and lower sufficiency indicate more faithful attributions.

| Method | Comprehensiveness | Sufficiency |
|---|---|---|
| Integrated Gradient | 0.6013 | 0.0290 |
| DeepLIFT | 0.3228 | 0.1573 |
| GIM | 0.5712 | 0.0392 |
| SHAP | 0.4800 | 0.1163 |
| LIME | 0.5220 | 0.1477 |
| Attn-Grad | **0.6031** | **−0.0081** |

## F.8. Uncertainty Quantification

Finally, we benchmark conformal prediction methods on TUEV at a target miscoverage level of $\alpha = 0.1$ (i.e., a target coverage of $0.9$). Results are averaged over random sample splits and reported as mean $\pm$ standard deviation.

*Table 22.* Conformal prediction on TUEV (Shah et al., 2018) ($\alpha = 0.1$, random sample splits). Empirical coverage and prediction set size are reported as mean $\pm$ standard deviation.

| Method | Empirical Coverage | Set Size |
|---|---|---|
| Conformal Prediction | $0.7461 \pm 0.0334$ | $1.11 \pm 0.21$ |
| + KDE Covariate Shift Adj. | $0.7457 \pm 0.0072$ | $1.11 \pm 0.14$ |
| + KMeans Adjustment | $0.7488 \pm 0.0152$ | $1.23 \pm 0.24$ |
| Neighborhood Conformal Prediction | $0.9152 \pm 0.0117$ | $1.25 \pm 0.13$ |

# G. PyHealth Datasets

PyHealth provides a comprehensive collection of healthcare datasets spanning multiple modalities and clinical domains. These datasets enable researchers to develop and evaluate machine learning models across diverse healthcare applications, from electronic health records to medical imaging, physiological signals, genomics, and clinical text. Each dataset is pre-processed and standardized to facilitate seamless integration with PyHealth's modeling and task frameworks.

**(a) Dataset Definition**

```python
class MIMIC3Dataset(BaseDataset):
    """
    A dataset class for handling MIMIC-III data.

    This class is responsible for loading and managing the MIMIC-III dataset,
    which includes tables such as patients, admissions, and icustays.

    Attributes:
        root (str): The root directory where the dataset is stored.
        tables (List[str]): A list of tables to be included in the dataset.
        dataset_name (Optional[str]): The name of the dataset.
        config_path (Optional[str]): The path to the configuration file.
    """

    def __init__(
        self,
        root: str,
        tables: List[str],
        dataset_name: Optional[str] = None,
        config_path: Optional[str] = None,
        **kwargs
    ) -> None:
        """
        Initializes the MIMIC4Dataset with the specified parameters.

        Args:
            root (str): The root directory where the dataset is stored.
            tables (List[str]): A list of additional tables to include.
            dataset_name (Optional[str]): The name of the dataset. Defaults to "mimic3".
            config_path (Optional[str]): The path to the configuration file. If not provided, a default config is used.
        """
        if config_path is None:
            logger.info("No config path provided, using default config")
            config_path = Path(__file__).parent / "configs" / "mimic3.yaml"
        default_tables = ["patients", "admissions", "icustays"]
        tables = default_tables + tables
        if "prescriptions" in tables:
            warnings.warn(
                "Events from prescriptions table only have date timestamp (no specific time). "
                "This may affect temporal ordering of events.",
                UserWarning,
            )
        super().__init__(
            root=root,
            tables=tables,
            dataset_name=dataset_name or "mimic3",
            config_path=config_path,
            **kwargs
        )
        return
```

**(b) Corresponding mimic3.yaml config**

```yaml
version: "1.4"
tables:
  patients:
    file_path: "PATIENTS.csv.gz"
    patient_id: "subject_id"
    timestamp: null
    attributes:
      - "gender"
      - "dob"
      - "dod"
      - "dod_hosp"
      - "dod_ssn"
      - "expire_flag"

  admissions:
    file_path: "ADMISSIONS.csv.gz"
    patient_id: "subject_id"
    timestamp: "admittime"
    attributes:
      - "hadm_id"
      - "admission_type"
      - "admission_location"
      - "insurance"
      - "language"
      - "religion"
      - "marital_status"
      - "ethnicity"
      - "edregtime"
      - "edouttime"
      - "diagnosis"
      - "discharge_location"
      - "dischtime"
      - "hospital_expire_flag"

  icustays:
    file_path: "ICUSTAYS.csv.gz"
    patient_id: "subject_id"
    timestamp: "intime"
    attributes:
      - "icustay_id"
      - "first_careunit"
      - "dbsource"
      - "last_careunit"
      - "outtime"

.... # More table definitions
```

*Figure 7.* **PyHealth 2.0 Dataset Example.** Defining a PyHealth dataset simply requires the inheritance of the (a) BaseDataset class and the definition of a (b) mimic3.yaml file for direct deployment.

## EHR Datasets

Electronic health record (EHR) datasets form the foundation of healthcare AI research, providing rich longitudinal patient data including diagnoses, procedures, medications, laboratory results, and clinical notes. PyHealth supports major publicly available EHR databases from intensive care units and broader hospital settings, along with standardized formats like OMOP that enable cross-institutional research.

## Medical Imaging Datasets

Medical imaging datasets enable the development of computer vision models for diagnostic support. PyHealth provides access to large-scale chest X-ray collections annotated with disease labels, supporting both single-label and multi-label

*Table 23.* Overview of PyHealth Dataset Categories

| Category | Count | Description |
|---|---|---|
| EHR Datasets | 7 | Electronic health records from ICU and hospital settings, including MIMIC-III/IV, eICU, and standardized OMOP formats |
| Medical Imaging | 3 | Chest X-ray datasets for disease classification and COVID-19 detection |
| Physiological Signals | 8 | EEG, ECG, and heart sound recordings for sleep staging, seizure detection, and cardiac monitoring |
| Genomics | 3 | Genetic variant and cancer mutation databases for pathogenicity prediction and survival analysis |
| Clinical Text | 1 | Medical transcription dataset for specialty classification |

*Table 24.* EHR Datasets in PyHealth

| Dataset | Description | Data Type | Key Features |
|---|---|---|---|
| MIMIC3Dataset | MIMIC-III Clinical Database v1.4 (Johnson et al., 2016) | EHR (ICU) | 46K+ patients, diagnoses, procedures, prescriptions, lab events, clinical notes |
| MIMIC4Dataset | MIMIC-IV Clinical Database v2.0+ (Johnson et al., 2023) | EHR (Hospital + ICU) | 300K+ patients, improved de-identification, expanded tables |
| eICUDataset | eICU Collaborative Research Database (Pollard et al., 2018) | EHR (ICU) | 200K+ ICU stays, 335 hospitals, diagnoses, medications, treatments |
| OMOPDataset | OMOP Common Data Model format (Hripcsak et al., 2015) | EHR (Standardized) | Supports any OMOP-formatted data |
| MIMICExtractDataset | MIMIC-Extract preprocessed benchmark (Wang et al., 2020) | EHR (Preprocessed) | Pre-processed features, standardized cohorts |
| EHRShotDataset | EHRShot benchmark dataset (Wornow et al., 2023) | EHR (Benchmark) | 15 predictive tasks |
| Support2Dataset | SUPPORT Study II survival data (Connors et al., 1995) | EHR (Survival) | 9,105 patients, survival outcomes |

classification tasks for thoracic pathologies and infectious diseases.

*Table 25.* Medical Imaging Datasets in PyHealth

| Dataset | Description | Data Type | Key Features |
|---|---|---|---|
| ChestXray14Dataset | NIH ChestX-ray14 Dataset (Wang et al., 2017) | Chest X-ray | 112K images, 14 disease labels |
| COVID19CXRDataset | COVID-19 Chest X-ray Dataset (Rahman et al., 2021) | Chest X-ray | COVID-19/Normal/Pneumonia classification |
| MIMICCXRDataset | MIMIC-CXR Chest X-ray Database (Johnson et al., 2019) | Chest X-ray | 377K images, linked to MIMIC-IV |

### Physiological Signal Datasets

Physiological signal datasets capture time-series recordings of brain activity (EEG), cardiac rhythms (ECG), and heart sounds (PCG). These datasets support the development of deep learning models for automated sleep staging, seizure detection, cardiac arrhythmia classification, and other diagnostic tasks that traditionally require expert manual annotation.

*Table 26.* Physiological Signal Datasets in PyHealth

| Dataset | Description | Data Type | Key Features |
|---|---|---|---|
| SleepEDFDataset | Sleep-EDF Database Expanded (Kemp et al., 2000) | EEG (Sleep) | 197 recordings, 5 sleep stages |
| ISRUCDataset | ISRUC-Sleep Dataset (Khalighi et al., 2016) | EEG (Sleep) | 100 subjects, 6-channel EEG, dual expert annotations |
| SHHSDataset | Sleep Heart Health Study (Quan et al., 1997) | EEG (Sleep) | 6K+ recordings, cardiovascular outcomes |
| TUABDataset | TUH Abnormal EEG Corpus (Obeid & Picone, 2016) | EEG (Clinical) | Abnormal vs normal EEG classification |
| TUEVDataset | TUH EEG Events Corpus (Shah et al., 2018) | EEG (Events) | EEG event detection, 6 event classes |
| CardiologyDataset (Perez Alday et al., 2022) | Cardiology ECG Dataset | ECG | Multiple arrhythmia types, 12-lead ECG |
| BMDHSDataset (Ali et al., 2024) | BMD Heart Sound Dataset | PCG (Audio) | Heart valve disease, 8 recordings/patient |
| DREAMTDataset (Wang et al., 2025a) | DREAMT Sleep Dataset | EEG (Sleep) | Multi-channel polysomnography |

### Genomics Datasets

Genomics datasets provide genetic and molecular information for precision medicine applications. These include variant databases with clinical significance annotations, somatic mutation catalogs from cancer patients, and multi-omics datasets linking genomic profiles to patient outcomes. Such datasets enable the development of models for pathogenicity prediction, mutation burden estimation, and survival analysis. From our team of PyHealth researchers, we contribute 3 different benchmarks that compose of our prostate cancer variant benchmark (Tavara et al., 2025).

*Table 27.* Genomics Datasets in PyHealth

| Dataset | Description | Data Type | Key Features |
| --- | --- | --- | --- |
| ClinVarDataset | ClinVar Genetic Variant Database (Landrum et al., 2018) | Variants | Clinical significance classification |
| COSMICDataset | COSMIC Cancer Mutation Database (Tate et al., 2019) | Mutations | Somatic mutations, FATHMM pathogenicity |
| TCGAPRADDataset | TCGA Prostate Adenocarcinoma (Weinstein et al., 2013) | Multi-omics | Mutations, clinical data, survival |

**Clinical Text Dataset**

Clinical text datasets contain unstructured medical narratives such as discharge summaries, radiology reports, and operative notes. These enable natural language processing research for medical specialty classification, named entity recognition, and clinical information extraction tasks.

*Table 28.* Clinical Text Dataset in PyHealth

| Dataset | Description | Data Type | Key Features |
| --- | --- | --- | --- |
| MedicalTranscriptionsDataset | MTSamples Medical Transcriptions | Text | Medical specialty classification |

## H. PyHealth Tasks

PyHealth provides a diverse collection of pre-defined clinical prediction tasks that transform raw healthcare data into machine learning-ready formats. These tasks implement standardized data preprocessing pipelines, feature extraction, and label generation for common healthcare prediction objectives. Each task is designed to work seamlessly with specific datasets and supports various model architectures through a unified interface.

*Table 29.* Overview of PyHealth Task Categories

| Task Category | Count | Description |
|---|---|---|
| Mortality Prediction | 9 | In-hospital and ICU mortality prediction across multiple EHR databases |
| Readmission Prediction | 4 | 30-day and general readmission risk prediction |
| Drug Recommendation | 2 | Medication combination recommendation with safety constraints |
| Length of Stay | 4 | Hospital length of stay classification into discrete bins |
| Sleep Staging | 4 | Automated sleep stage classification from EEG signals |
| EEG Analysis | 2 | Abnormality detection and event classification |
| Medical Imaging | 3 | Disease classification from chest X-rays |
| Cardiology Detection | 5 | ECG-based cardiac condition identification |
| Genomics/Cancer | 4 | Variant pathogenicity and cancer survival prediction |
| Other Specialized | 6 | Benchmark suites, medical coding, and patient linkage |

### Mortality Prediction Tasks

Mortality prediction tasks aim to identify patients at high risk of death during hospitalization or within a specified timeframe. These tasks are critical for clinical decision support, resource allocation, and early intervention. PyHealth supports mortality prediction across multiple EHR databases with varying feature representations, including structured codes, laboratory values, and multimodal data incorporating clinical notes and medical images.

*Table 30.* Mortality Prediction Tasks in PyHealth

| Task | Dataset | Output | Input Features |
|---|---|---|---|
| MortalityPredictionMIMIC3 | MIMIC-III | Binary | conditions, procedures, drugs |
| MortalityPredictionMIMIC4 | MIMIC-IV | Binary | conditions, procedures, drugs |
| MultimodalMortalityPrediction-MIMIC3 | MIMIC-III | Binary | conditions, procedures, drugs, clinical_notes |
| MultimodalMortalityPrediction-MIMIC4 | MIMIC-IV | Binary | conditions, procedures, drugs, discharge, radiology, labs, image |
| MortalityPredictionEICU | eICU | Binary | conditions, procedures, drugs |
| MortalityPredictionEICU2 | eICU | Binary | conditions (admissionDx), procedures (treatment) |
| MortalityPredictionOMOP | OMOP | Binary | conditions, procedures, drugs |
| MortalityPrediction-StageNetMIMIC4 | MIMIC-IV | Binary | icd_codes (stagenet), labs (tensor) |
| InHospitalMortalityMIMIC4 | MIMIC-IV | Binary | labs (timeseries) |

### Readmission Prediction Tasks

Hospital readmission prediction identifies patients likely to return to the hospital within a specified period (commonly 30 days) after discharge. Reducing preventable readmissions improves patient outcomes and reduces healthcare costs. These

tasks leverage comprehensive patient histories including diagnoses, procedures, and medications to assess readmission risk.

*Table 31.* Readmission Prediction Tasks in PyHealth

| Task | Dataset | Output | Input Features |
| --- | --- | --- | --- |
| ReadmissionPredictionMIMIC3 | MIMIC-III | Binary | conditions, procedures, drugs |
| Readmission30DaysMIMIC4 | MIMIC-IV | Binary | conditions, procedures, drugs |
| readmission_prediction_eicu_fn | eICU | Binary | conditions, procedures, drugs |
| ReadmissionPredictionOMOP | OMOP | Binary | conditions, procedures, drugs |

**Drug Recommendation Tasks**

Drug recommendation tasks predict optimal medication combinations for patients based on their medical history and current conditions. These tasks are particularly challenging due to the need to avoid harmful drug-drug interactions (DDI) while maximizing therapeutic efficacy. The output is typically a multilabel prediction where multiple medications can be recommended simultaneously.

*Table 32.* Drug Recommendation Tasks in PyHealth

| Task | Dataset | Output | Input Features |
| --- | --- | --- | --- |
| DrugRecommendationMIMIC3 | MIMIC-III | Multilabel | conditions, procedures, drugs_hist (nested) |
| DrugRecommendationMIMIC4 | MIMIC-IV | Multilabel | conditions, procedures, drugs_hist (nested) |

**Length of Stay Prediction Tasks**

Length of stay (LOS) prediction estimates how long a patient will remain hospitalized, typically discretized into clinically meaningful bins (e.g., 0-1 days, 1-2 days, etc.). Accurate LOS prediction supports hospital resource planning, bed management, and discharge planning. These tasks frame the problem as multiclass classification with 10 ordinal categories.

*Table 33.* Length of Stay Prediction Tasks in PyHealth

| Task | Dataset | Output | Input Features |
| --- | --- | --- | --- |
| LengthOfStayPredictionMIMIC3 | MIMIC-III | Multiclass (10) | conditions, procedures, drugs |
| LengthOfStayPredictionMIMIC4 | MIMIC-IV | Multiclass (10) | conditions, procedures, drugs |
| LengthOfStayPredictioneICU | eICU | Multiclass (10) | conditions, procedures, drugs |
| LengthOfStayPredictionOMOP | OMOP | Multiclass (10) | conditions, procedures, drugs |

**Sleep Staging Tasks**

Sleep staging tasks classify EEG recordings into distinct sleep stages (Wake, N1, N2, N3, REM) following standard polysomnography scoring criteria. Automated sleep staging reduces the burden of manual annotation by sleep technologists and enables large-scale sleep research. These tasks process multi-channel EEG signals segmented into 30-second epochs.

**EEG Analysis Tasks**

EEG analysis tasks focus on detecting abnormalities and specific events in electroencephalography recordings. Abnormality detection classifies entire EEG recordings as normal or abnormal, supporting clinical screening workflows. Event detection identifies transient patterns such as seizures, spikes, and other neurological phenomena that require medical attention.

*Table 34.* Sleep Staging Tasks in PyHealth

| Task | Dataset | Output | Input Features |
|---|---|---|---|
| sleep_staging_sleepedf_fn | SleepEDF | Multiclass (5-6) | EEG signal epochs |
| sleep_staging_isruc_fn | ISRUC | Multiclass (5) | EEG signal epochs |
| sleep_staging_shhs_fn | SHHS | Multiclass (5) | EEG signal epochs |
| SleepStagingSleepEDF | SleepEDF | Multiclass | EEG signal (class-based) |

*Table 35.* EEG Analysis Tasks in PyHealth

| Task | Dataset | Output | Input Features |
|---|---|---|---|
| EEG_isAbnormal_fn | TUAB | Binary | 16-channel bipolar EEG |
| EEGEventsTUEV | TUEV | Multiclass (6) | 16-channel bipolar EEG |

## Medical Imaging Tasks

Medical imaging tasks apply computer vision techniques to diagnostic images, particularly chest X-rays. These tasks range from binary classification of single pathologies to multilabel classification where multiple conditions may be present simultaneously. They support radiologist workflow augmentation and automated screening in resource-limited settings.

*Table 36.* Medical Imaging Tasks in PyHealth

| Task | Dataset | Output | Input Features |
|---|---|---|---|
| ChestXray14BinaryClassification | ChestXray14 | Binary | X-ray image |
| ChestXray14MultilabelClassification | ChestXray14 | Multilabel (14) | X-ray image |
| COVID19CXRClassification | COVID19CXR | Multiclass (3) | X-ray image |

## Cardiology Detection Tasks

Cardiology detection tasks identify specific cardiac conditions from ECG recordings. Each task targets a clinically relevant condition such as valve disease, conduction disorders, or arrhythmias. These binary classification tasks enable automated ECG interpretation and can serve as diagnostic support tools or screening mechanisms in primary care settings.

## Genomics and Cancer Tasks

Genomics and cancer tasks predict clinically relevant outcomes from genetic and molecular data. These include classifying the pathogenicity of genetic variants, estimating tumor mutation burden, and predicting patient survival based on genomic profiles. Such tasks advance precision oncology by enabling risk stratification and treatment selection based on molecular characteristics.

## Other Specialized Tasks

Additional specialized tasks address diverse healthcare applications including benchmark evaluations, medical coding from clinical notes, patient record linkage, and heart sound classification. These tasks demonstrate PyHealth's versatility across different healthcare informatics problems beyond traditional clinical prediction.

*Table 37.* Cardiology Detection Tasks in PyHealth

| Task | Dataset | Output | Condition Detected |
|---|---|---|---|
| cardiology_isAR_fn | Cardiology | Binary | Aortic Regurgitation |
| cardiology_isBBBFB_fn | Cardiology | Binary | Bundle Branch Block / Fascicular Block |
| cardiology_isAD_fn | Cardiology | Binary | Atrial Disorders |
| cardiology_isCD_fn | Cardiology | Binary | Conduction Disorders |
| cardiology_isWA_fn | Cardiology | Binary | Wolf-Parkinson-White / Arrhythmias |

*Table 38.* Genomics and Cancer Tasks in PyHealth

| Task | Dataset | Output | Input Features |
|---|---|---|---|
| CancerSurvivalPrediction | TCGA-PRAD | Binary | mutations, age, Gleason score |
| CancerMutationBurden | TCGA-PRAD | Binary | mutations, age |
| VariantClassificationClinVar | ClinVar | Multiclass (5) | gene_symbol, variant_type, chromosome |
| MutationPathogenicityPrediction | COSMIC | Binary | gene_name, mutation_description, primary_site |

*Table 39.* Other Specialized Tasks in PyHealth

| Task | Dataset | Output | Description |
|---|---|---|---|
| BenchmarkEHRShot | EHRShot | Various | 15 benchmark tasks |
| BMDHSDiseaseClassification | BMD-HS | Multilabel (4) | Heart valve disease (AS, AR, MR, MS) |
| SurvivalPreprocessSupport2 | Support2 | Survival | Survival outcome preprocessing |
| MedicalTranscriptionsClassification | MTSamples | Multiclass | Medical specialty from transcription |
| MIMIC3ICD9Coding | MIMIC-III | Multilabel | ICD-9 code prediction from notes |
| patient_linkage_mimic3_fn | MIMIC-III | Linkage | Patient record linkage |

# I. PyHealth Models

PyHealth implements a comprehensive collection of machine learning and deep learning models tailored for healthcare applications. The model library spans general-purpose sequence models adapted for medical data, healthcare-specific architectures designed for clinical prediction tasks, specialized drug recommendation models that incorporate pharmacological knowledge, and auxiliary models for generation, graph learning, and transfer learning. All models follow a unified API that enables consistent training, evaluation, and deployment workflows.

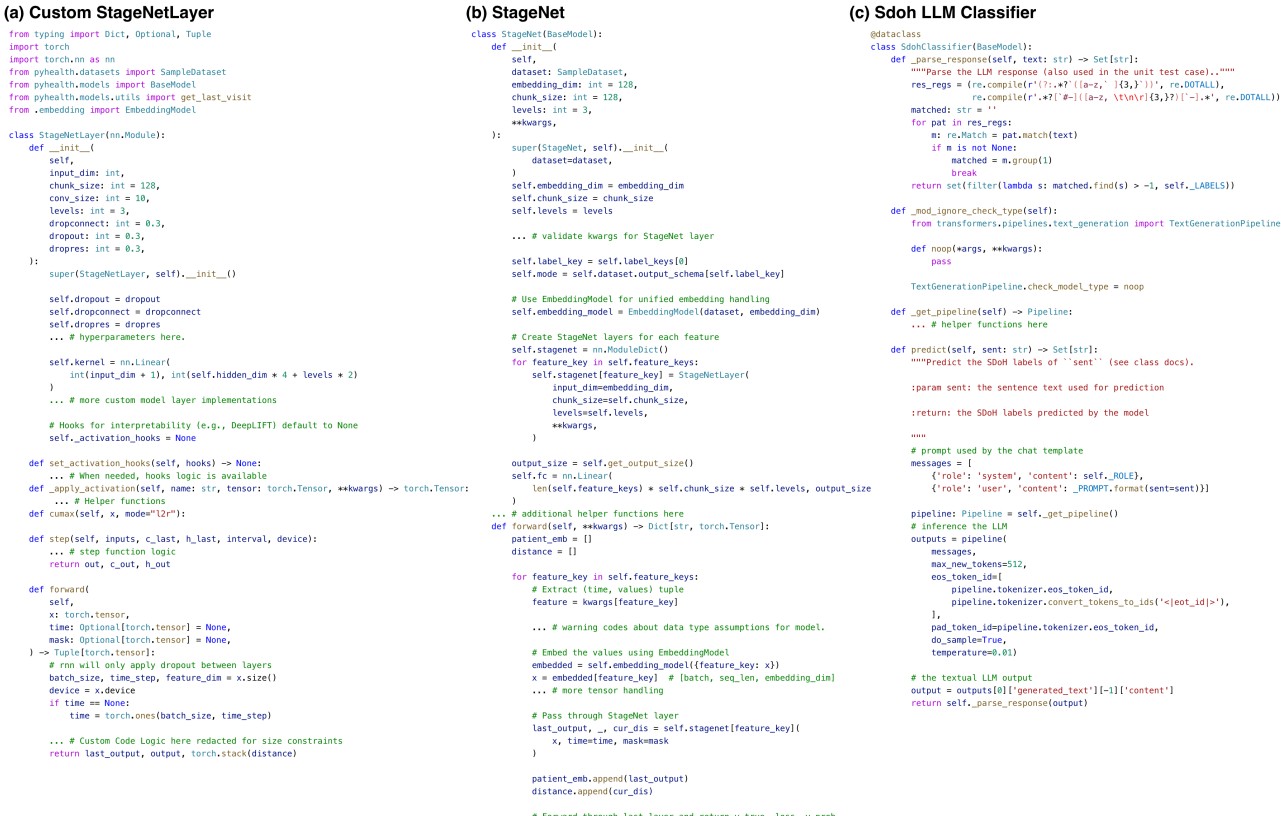

*Figure 8.* **PyHealth 2.0 Models.** PyHealth offers examples of modular model layers like the StageNetLayer (a) as well as custom models like StageNet (b), and even LLMs for social determinants of health extraction (c) (Guevara et al., 2024; Landes et al., 2025). In this case, PyHealth's model class makes very little assumptions on what's allowed for model usage beyond requirements in documentation as shown by the LLM wrapper (c). However, for direct use with pre-existing dataloaders and the built-in trainer, we highly recommend users to follow a more strict forward() operation where users are recommended to return the predictions and loss functions of their models (b) and more closely follow schema requirements made from PyHealth tasks here.

### General Sequence Models

General sequence models provide foundational architectures for processing temporal healthcare data. These models can be applied across different prediction tasks and data modalities. Recurrent neural networks (RNNs) capture temporal dependencies in patient visits, Transformers leverage attention mechanisms for long-range interactions, convolutional neural networks (CNNs) extract local patterns from signals and images, temporal convolutional networks (TCNs) offer efficient sequence modeling, and multi-layer perceptrons (MLPs) serve as baseline classifiers.

### Healthcare-Specific Models

Healthcare-specific models incorporate domain knowledge and specialized architectures designed for clinical data characteristics. RETAIN provides interpretable predictions through reverse-time attention, StageNet captures disease progression stages, AdaCare and ConCare adapt feature importance based on patient context, GRASP leverages patient similarity graphs, Agent employs multi-agent reinforcement learning, Deepr applies CNNs to medical record sequences, and SparcNet and

*Table 40.* Overview of PyHealth Model Categories

| Model Category | Count | Description |
| --- | --- | --- |
| General Sequence Models | 5 | RNN, Transformer, CNN, TCN, and MLP architectures for general healthcare prediction |
| Healthcare-Specific | 10 | Models designed specifically for EHR and clinical data including RETAIN, StageNet, AdaCare, and ConCare |
| Drug Recommendation | 4 | Specialized models for safe medication combination recommendation with DDI constraints |
| Specialized Models | 9 | Graph neural networks, generative models, LLM-based classifiers, and pre-trained vision/language models |

*Table 41.* General Sequence Models in PyHealth

| Model | Architecture | Task Types | Reference |
| --- | --- | --- | --- |
| RNN | Recurrent Neural Network (GRU/LSTM/Vanilla) | Classification, Regression | (Hochreiter & Schmidhuber, 1997; Chung et al., 2014) |
| Transformer | Self-attention based encoder | Classification, Regression | (Vaswani et al., 2017) |
| CNN | Convolutional Neural Network | Signal/Image tasks | (LeCun et al., 2002) |
| TCN | Temporal Convolutional Network | Sequence tasks | (Bai et al., 2018) |
| (Rumelhart et al., 1986) | | | |

ContraWR handle physiological signals with sparse convolutions and contrastive learning respectively.

**Drug Recommendation Models**

Drug recommendation models address the complex problem of suggesting safe and effective medication combinations. SafeDrug incorporates drug-drug interaction (DDI) knowledge graphs with dual molecular encoders, GAMENet uses graph-augmented memory networks to capture medication dependencies, MICRON models medication changes with recurrent residual connections, and MoleRec leverages molecular substructure representations for improved generalization across drug vocabularies.

**Specialized Models**

Specialized models extend PyHealth's capabilities to additional domains and methodologies. SdohClassifier applies large language models with LoRA fine-tuning for social determinants of health extraction, MedLink performs patient record linkage, VAE and GAN enable generative modeling, GAT and GCN provide graph neural network implementations, TorchvisionModel wraps pre-trained computer vision models, and TransformersModel integrates HuggingFace language models including domain-specific variants like ClinicalBERT.

*Table 42.* Healthcare-Specific Models in PyHealth

| Model | Description | Task Types | Reference |
|---|---|---|---|
| RETAIN | Reverse Time Attention Model with interpretable attention | Prediction | (Choi et al., 2016) |
| StageNet | Stage-aware neural network for health risk prediction | Prediction | (Gao et al., 2020b) |
| StageAttentionNet | StageNet with Multi-Head Attention between SA-LSTM and CNN | Prediction | Extended StageNet |
| AdaCare | Adaptive feature importance calibration for EHR | Prediction | (Ma et al., 2020a) |
| ConCare | Context-aware health status representation learning | Prediction | (Ma et al., 2020b) |
| GRASP | Graph-based patient similarity with clustering | Prediction | (Zhang et al., 2021) |
| Agent | Dr. Agent - Multi-agent RL with dynamic skip connections | Prediction | (Gao et al., 2020a) |
| Deepr | CNN for medical records with max pooling | Prediction | (Nguyen et al., 2016) |
| SparcNet | Sparse CNN for EEG signals | Signal | (Jing et al., 2023) |
| ContraWR | Contrastive learning for waveform recognition | Signal | (Yang et al., 2021c) |

*Table 43.* Drug Recommendation Models in PyHealth

| Model | Description | Task Type | Reference |
|---|---|---|---|
| SafeDrug | Safe drug recommendation with DDI knowledge graph | Drug Rec. | (Yang et al., 2021b) |
| GAMENet | Graph Augmented Memory Network for drug recommendation | Drug Rec. | (Shang et al., 2019) |
| MICRON | Change-aware drug recommendation with DDI | Drug Rec. | (Yang et al., 2021a) |
| MoleRec | Molecular structure-aware drug recommendation | Drug Rec. | (Yang et al., 2023b) |

*Table 44.* Specialized Models in PyHealth

| Model | Description | Task Type | Reference |
|---|---|---|---|
| SdohClassifier | LLM-based Social Determinants of Health classifier (Llama 3.1 + LoRA) | NLP | (Guevara et al., 2024) |
| MedLink | Patient linkage across EHR systems | Linkage | Patient record matching |
| LogisticRegression | Logistic regression baseline | Classification | Standard baseline |
| VAE | Variational Autoencoder | Generation | (Kingma & Welling, 2013) |
| GAN | Generative Adversarial Network | Generation | (Goodfellow et al., 2014) |
| GAT | Graph Attention Network | Graph | (Veličković et al., 2017) |
| GCN | Graph Convolutional Network | Graph | (Kipf, 2016) |
| TorchvisionModel | Pretrained vision models (ResNet, etc.) | Image | torchvision models |
| TransformersModel | HuggingFace Transformers integration | NLP/Multi | (Devlin et al., 2019; Alsentzer et al., 2019) |

## J. PyHealth Data Processors

PyHealth provides a modular system of data processors that transform raw healthcare data into model-ready representations. These processors handle diverse data modalities including sequences, images, signals, text, and structured tabular data. Each processor implements standardized interfaces for fitting vocabularies, encoding features, and batching samples, enabling seamless integration between datasets and models regardless of the underlying data format.

*Table 45.* Overview of PyHealth Processor Categories

| Processor Category | Count | Description |
| --- | --- | --- |
| Sequence Processors | 5 | Handle temporal sequences of medical codes and nested structures |
| Signal & Image | 3 | Process physiological signals, audio, and medical images |
| Label Processors | 4 | Encode prediction targets for classification and regression tasks |
| Specialized Processors | 5 | Domain-specific processors for StageNet, multi-hot encoding, and raw data |

### Sequence Processors

Sequence processors handle temporal medical data including visit sequences, medication histories, and longitudinal patient records. These processors build vocabularies from medical codes (diagnoses, procedures, drugs), perform tokenization, and create padded sequences suitable for recurrent and attention-based models. Nested processors handle hierarchical structures where each visit contains multiple codes, while deep nested processors support additional nesting levels for complex data representations.

*Table 46.* Sequence Processors in PyHealth

| Processor | Description | Use Cases |
| --- | --- | --- |
| SequenceProcessor | Processes flat sequences of tokens/codes | Time series of single values, medication sequences |
| NestedSequenceProcessor | Handles sequences of code sets (visits) | EHR visit sequences with multiple codes per visit |
| NestedFloatsProcessor | Processes nested sequences with numerical values | Laboratory values, vital signs over visits |
| DeepNestedSequenceProcessor | Three-level nesting for complex hierarchies | Drug recommendation with historical medication sets |
| DeepNestedFloatsProcessor | Three-level nesting with float values | Multi-visit laboratory panels with multiple tests |

### Signal and Image Processors

Signal and image processors prepare continuous waveforms and medical images for deep learning models. Signal processors handle EEG, ECG, and other physiological time series through resampling, segmentation, and normalization. Image processors load medical images, apply preprocessing transforms, and ensure consistent tensor formatting. Audio processors specifically handle phonocardiogram (heart sound) data with domain-specific preprocessing.

### Label Processors

Label processors encode prediction targets into appropriate formats for model training. They handle diverse output types including binary classification, multiclass problems with softmax, multilabel tasks where multiple labels can be active simultaneously, and continuous regression targets. These processors ensure consistent label representations across different tasks and datasets.

*Table 47.* Signal and Image Processors in PyHealth

| Processor | Description | Use Cases |
|---|---|---|
| SignalProcessor | Processes 1D continuous signals | EEG, ECG, PPG waveforms |
| AudioProcessor | Handles audio waveforms with specialized transforms | Heart sounds, respiratory sounds |
| ImageProcessor | Loads and preprocesses medical images | Chest X-rays, CT scans, pathology slides |

*Table 48.* Label Processors in PyHealth

| Processor | Description | Output Format |
|---|---|---|
| BinaryLabelProcessor | Binary classification labels | Single probability |
| MultiClassLabelProcessor | Mutually exclusive class labels | Class index or one-hot vector |
| MultiLabelProcessor | Multiple simultaneous labels | Binary vector for each class |
| RegressionLabelProcessor | Continuous target values | Scalar or vector of floats |

**Specialized Processors**

Specialized processors address unique requirements of specific models or data formats. StageNet processors create tensor representations with stage-aware features for disease progression modeling. Multi-hot processors encode presence/absence patterns efficiently. Timeseries processors handle irregular temporal data with timestamps. Text processors prepare clinical notes and reports for language models, while raw processors pass data through unchanged for custom handling.

*Table 49.* Specialized Processors in PyHealth

| Processor | Description | Use Cases |
|---|---|---|
| StageNetProcessor | Stage-aware ICD code processing | StageNet model with disease progression |
| StageNetTensorProcessor | Tensor representation for StageNet | Laboratory values with stage awareness |
| MultiHotProcessor | Binary vector encoding of presence | Efficient code set representation |
| TimeseriesProcessor | Irregular time series with timestamps | Vital signs, lab values with variable sampling |
| TextProcessor | Clinical text preprocessing | Discharge summaries, radiology reports |
| TensorProcessor | Generic tensor handling | Pre-processed numerical features |
| RawProcessor | Pass-through without transformation | Custom preprocessing pipelines |
| IgnoreProcessor | Placeholder for unused fields | Excluding fields from processing |

## K. PyHealth Interpretability Methods

PyHealth implements a comprehensive suite of interpretability methods that explain model predictions through feature attribution and visualization techniques. These methods help clinicians and researchers understand which input features (e.g., specific diagnoses, lab values, or image regions) contribute most to predictions, enabling model validation, bias detection, and clinical insight discovery. The interpretability module supports both gradient-based and perturbation-based approaches across different data modalities, addressing the critical need for explainable AI in healthcare applications (Cinà et al., 2025).

**Gradient-Based Attribution Methods**

Gradient-based methods compute feature importance by analyzing how model outputs change with respect to input features (Simonyan et al., 2013). These methods are computationally efficient and work well with differentiable models. Basic

*Table 50.* Overview of PyHealth Interpretability Methods

| Method Category | Count | Description |
|---|---|---|
| Gradient-Based | 4 | Attribution through backpropagation of gradients |
| Perturbation-Based | 2 | Attribution through input perturbations |
| Attention-Based | 1 | Attribution from transformer attention mechanisms |
| Visualization Tools | 4+ | Utilities for displaying and overlaying attributions |

gradient saliency maps provide first-order approximations, while integrated gradients (Sundararajan et al., 2017) and DeepLift (Shrikumar et al., 2017) offer more sophisticated attributions that satisfy desirable theoretical properties like completeness and sensitivity.

*Table 51.* Gradient-Based Interpretability Methods in PyHealth

| Method | Description | Key Properties |
|---|---|---|
| BasicGradientSaliencyMaps (Simonyan et al., 2013) | Computes input gradient magnitude | Fast, first-order approximation |
| IntegratedGradients (Sundararajan et al., 2017) | Path integral of gradients from baseline | Satisfies completeness axiom |
| DeepLift (Shrikumar et al., 2017) | Backpropagates contributions relative to reference | Handles saturation, efficient |
| GIM (Gradient Input Multiplication) (Edin et al., 2025) | Element-wise product of gradients and inputs | Highlights salient input regions |

**Perturbation-Based Attribution Methods**

Perturbation-based methods assess feature importance by observing how predictions change when inputs are masked or modified. LIME (Ribeiro et al., 2016) builds local linear approximations around individual predictions, while SHAP (Lundberg & Lee, 2017) computes Shapley values that provide game-theoretic optimal attributions. These methods are model-agnostic and can provide more faithful explanations at the cost of increased computational requirements.

*Table 52.* Perturbation-Based Interpretability Methods in PyHealth

| Method | Description | Key Properties |
|---|---|---|
| LimeExplainer | Local Interpretable Model-agnostic Explanations | Model-agnostic, local fidelity |
| ShapExplainer | Shapley Additive exPlanations values | Theoretically optimal, consistent |

**Attention-Based Attribution Methods**

Attention-based methods leverage built-in attention mechanisms in transformer models to derive feature importance. Chefer relevance propagation specifically addresses how to properly propagate relevance through multi-layer transformers, combining attention weights with gradient information to provide accurate attributions for vision transformers and other attention-based architectures.

# L. PyHealth Uncertainty Quantification

PyHealth provides post-hoc uncertainty quantification methods that improve the reliability and trustworthiness of model predictions (He et al., 2025). The calibration module addresses two complementary aspects: probability calibration adjusts

*Table 53.* Attention-Based Interpretability Methods in PyHealth

| Method | Description | Key Properties |
|---|---|---|
| CheferRelevance (Chefer et al., 2021) | Transformer-specific relevance propagation | Handles multi-head attention, layer propagation |

predicted probabilities to better reflect true confidence levels, while prediction set methods construct set-valued predictions with statistical coverage guarantees. These techniques are crucial for deploying healthcare AI systems in clinical settings where miscalibration can lead to harmful decisions.

*Table 54.* Overview of PyHealth Uncertainty Quantification Methods

| Method Category | Count | Description |
|---|---|---|
| Calibration Methods | 4 | Post-hoc probability calibration techniques |
| Prediction Sets | 4 | Conformal prediction and set-valued classifiers |

**Probability Calibration Methods**

Probability calibration methods adjust model outputs to ensure that predicted probabilities accurately reflect empirical frequencies (Guo et al., 2017). Temperature scaling learns a single scalar parameter to recalibrate logits, histogram binning uses non-parametric binning strategies (Zadrozny & Elkan, 2001), Dirichlet calibration employs matrix transformations with regularization (Kull et al., 2019), and KCal leverages kernel density estimation on learned embeddings for multiclass calibration (Lin et al., 2022b). These methods require a held-out calibration set and can significantly improve clinical decision-making based on predicted probabilities.

*Table 55.* Probability Calibration Methods in PyHealth

| Method | Description | Modes | Reference |
|---|---|---|---|
| TemperatureScaling | Scalar temperature parameter for logit scaling | binary, multiclass, multilabel | (Guo et al., 2017) |
| HistogramBinning | Non-parametric binning of predictions | binary, multiclass, multilabel | (Zadrozny & Elkan, 2001) |
| DirichletCalibration | Matrix transformation with regularization | multiclass | (Kull et al., 2019) |
| KCal | Kernel density estimation on embeddings | multiclass | (Lin et al., 2022b) |

**Prediction Set Methods**

Prediction set methods provide set-valued predictions with statistical guarantees on coverage or error rates. Instead of outputting a single class, these methods return a set of plausible classes that contains the true label with high probability. LABEL (Sadinle et al., 2019) and SCRIB (Lin et al., 2022a) offer complementary approaches with marginal and class-conditional coverage, FavMac (Lin et al., 2023) handles multilabel scenarios with cost control, and CovariateLabel (Tibshirani et al., 2019; Laghuvarapu et al., 2023) extends conformal prediction to handle distribution shift. These methods enable more conservative but reliable predictions in high-stakes clinical applications.

*Table 56.* Prediction Set Methods in PyHealth

| Method | Description | Modes | Reference |
|---|---|---|---|
| LABEL | Least Ambiguous set-valued classifier | multiclass | (Sadinle et al., 2019) |
| SCRIB | Class-specific risk bounds with optimized thresholds | multiclass | (Lin et al., 2022a) |
| FavMac | Value-maximizing sets with cost control | multilabel | (Lin et al., 2023) |
| CovariateLabel | Conformal prediction under covariate shift | multiclass | (Tibshirani et al., 2019; Laghuvarapu et al., 2023) |

