# OpenReview forum: "PyHealth 2.0: A Comprehensive Open-Source Toolkit for Accessible and Reproducible Clinical Deep Learning"
_ICML.cc/2026/Conference — ICML 2026 regular_

### Official Review · Reviewer_qwws · 2026-03-02

**Soundness:** 1
**Presentation:** 3
**Significance:** 2
**Originality:** 1
**Overall Recommendation:** 2
**Confidence:** 4

**Summary:**

This manuscript introduces PyHealth 2.0, an open-source clinical deep learning toolkit engineered to overcome persistent challenges in reproducibility, computational overhead, and specialized domain expertise. The authors present a unified framework that integrates over 15 datasets, 20 tasks, and 25 models. By supporting diverse data modalities (signals, text, imaging, EHR), the toolkit prioritizes accessibility for clinical researchers. Furthermore, by leveraging Dask and Polars for lazy loading and multi-worker sharding, the framework enables high-performance, large-scale clinical modeling on consumer-grade hardware. The authors claim that these optimizations result in up to $39\times$ faster task processing and $20\times$ lower memory usage compared to previous versions.

**Compliance With Llm Reviewing Policy:**

Affirmed.

**Final Justification:**

I keep my original score. The authors' response did not convince me, and I still have the same concerns.

**Key Questions For Authors:**

1- Could you provide a stress test or benchmark conducted on a physical 16GB machine to demonstrate that the wall-clock time and swap-space usage remain viable for a researcher without access to a workstation?

2- According to Table 3, PyHealth 1.16 is faster than version 2.0 in wall-clock time for the Mortality Prediction task across almost all worker counts (e.g., 6,841s vs. 17,671s at 1 worker). Since the paper identifies "high computational costs" as a persistent barrier, how do you justify a new version that is significantly slower in execution time for its most memory-intensive task?

3- While the toolkit claims to support 15+ datasets, including signals, genomics, and imaging, the performance metrics (Wall Time and RAM) are only reported for EHR-based tasks using the MIMIC-IV dataset. To substantiate the claim of a "unified multimodal framework," can you provide performance benchmarks for non-EHR modalities?

**Limitations:**

- The authors should explicitly acknowledge that PyHealth 2.0 exhibits a performance regression in wall time for the Mortality Prediction task compared to version 1.16. This is a significant technical limitation that contradicts the general claim of "faster processing" and should be documented to help users choose the right version for their needs.

- The manuscript claims to support 16GB laptops, yet all benchmarks were conducted on a workstation with 1TB of RAM. The authors should list the lack of empirical testing on resource-constrained hardware as a limitation, as the real-world performance on a consumer laptop (where disk I/O and CPU throttling are factors) remains unverified.

- The limitation section should state that the toolkit’s performance has only been quantitatively verified for EHR modalities. Users should be cautioned that the multimodal claims for imaging, genomics, and biosignals currently lack the same level of rigorous performance benchmarking.

**Strengths And Weaknesses:**

# Soundness

The empirical evaluation utilizes the MIMIC-IV dataset, encompassing over 300,000 patients and millions of clinical events. However, the benchmarking methodology raises concerns regarding fairness; comparing a multi-threaded, C++/Rust-backed library (Polars/Dask) against an unoptimized, single-worker Pandas baseline may not represent a rigorous standard. Established clinical ML researchers typically move beyond unoptimized Pandas for large-scale EHR joins, suggesting that a more robust comparison would include high-performance SQL-based pipelines or frameworks outside the MEDS ecosystem.

Furthermore, the results indicate performance regressions; in the Mortality Prediction task (Table 3), PyHealth 1.16 actually outperforms version 2.0 in wall time. While the authors posit that the system enables modeling on 16GB laptops, all reported benchmarks were executed on a high-end workstation equipped with 1TB of RAM and a 32-core processor. Although PyHealth 2.0’s RAM consumption remains below the 16GB threshold, the manuscript lacks empirical evidence of performance on truly resource-constrained hardware.

The absence of significance tests and missing performance metrics for non-EHR modalities, such as biosignals, genomics, and imaging, further weakens these technical claims. To fully validate the toolkit’s versatility, the evaluation requires a more diversified experimental setup spanning multiple modalities and datasets across a broader range of models.

# Presentation

The paper is well-structured. The transition from the reproducibility crisis to specific modular solutions is logically consistent, and diagrams are instructive. The claim of "7 lines of code" feels more like marketing than a technical metric.

# Significance

By lowering the hardware requirement to 16GB RAM, the authors help democratize clinical AI research for researchers without large compute clusters. While the toolkit is a useful resource, its impact on methodological advancement is limited. It is primarily an engineering optimization of existing methods. It wraps complex experimental details into high-level functions and uses C++/Rust-backed libraries, a common trend in the Python ecosystem, to improve performance.

# Originality
Allowing researchers to combine imaging and EHR data without the heavy dependencies common in frameworks like MONAI or Zensols is a step forward. However, the core architectural components (Polars, Dask, PyTorch Lightning) are existing industry standards. The paper’s originality lies in the integration of these tools for a specific healthcare vertical rather than a novel technical contribution.

---

> ### Author Rebuttal · Authors · 2026-03-29
>
> * **[Laptop benchmark]** We apologize for omitting these results. We benchmarked the full `labevents` mortality task on a MacBook Pro M2 Pro (16GB) and an ASUS Zephyrus G16 (32GB DDR5, WSL). Both outperform our workstation due to: (1) NAS-based I/O bottlenecks on the workstation vs. local NVMe SSDs on laptops; (2) higher memory bandwidth on both laptops vs. DDR4; and (3) higher single-core performance, which is the relevant metric for our sequential streaming workload. We will revise Figure 1 to include these results and note in the limitations that performance may vary across hardware configurations.
>
> **MIMIC-IV `labevents` Mortality — Data Processing Benchmarks (`num_workers = 8`)**
>
> | Device | Wall Time (s) | Peak Mem (GB) |
> | :----- | ----: | ----: |
> | Large Workstation (NAS, DDR4) | 2,975 | 23.57 |
> | MacBook Pro M2 Pro (16GB) | 1,357 | 7.43 |
> | ASUS Zephyrus G16 (32GB DDR5) | 1,604.65 | 19.67 |
> | Pandas (baseline) | 93,708 | 49.23 |
>
> **(`num_workers = 4`)**
>
> | Device | Wall Time (s) | Peak Mem (GB) |
> | :----- | ----: | ----: |
> | Large Workstation (NAS, DDR4) | 4,605 | 23.39 |
> | MacBook Pro M2 Pro (16GB) | 2,009.53 | 5.31 |
> | ASUS Zephyrus G16 (32GB DDR5) | 2,617.41 | 19.15 |
> | Pandas (baseline) | 93,708 | 49.23 |
>
> ---
>
> * **[Wall-clock regression vs. PyHealth 1.16]** The slower single-worker performance is a direct and intentional consequence of PyHealth 2.0's memory design: rather than loading the entire dataset into memory, PyHealth 2.0 writes processed data to disk and lazily iterates over it, reading only what is needed at any given time. This is what makes consumer hardware viable—PyHealth 1.16's multi-worker setup requires holding the full dataset in RAM, which exceeds the capacity of most machines. At 4 workers, PyHealth 2.0 is already comparable to 1.16 in wall-clock time. At 8 workers, the MacBook Pro achieves 1,357s—faster than PyHealth 1.16 at 16 workers (2,235s)—while using a fraction of the memory. Modern consumer hardware routinely provides 4+ cores, so this advantage is accessible without a workstation. We will revise our wording around when PyHealth 2.0 is superior here.
>
> ---
>
> * **[Non-EHR modality performance]** Our processing benchmarks in the paper were EHR-only. Below we provide data processing benchmarks (wall time = loading + task processing, peak memory) for image, text, and biosignal modalities. Downstream model performance for these modalities are in our responses to Reviewers 8Zsd and ZM1B. Regarding dataset sizes, TUAB contains 409,555 EEG samples; TUEV contains 112,491; CovidCXR contains ~6,000 X-rays; Medical Transcriptions contains 4,998 records. As a key note, this processing includes converting specialized files to torch tensors.
>
> | Modality | Dataset | Wall Time (s) | Peak Mem (GB) |
> | :------- | :------ | ------------: | ---------------: |
> | Image | Covid19CXR | 142.79 | 1.95 |
> | Text | Medical Transcriptions | 48.46 | 2.24 |
> | Signals (EEG) | TUAB | 6,547.78 | 2.05 |
> | Signals (EEG) | TUEV | 1,210.79 | 2.78 |
>
> ---
>
> * **[Originality / integration contribution]** We agree the originality lies in vertical integration rather than a novel algorithm. We push back, however, on the framing that combining Dask, Polars, and PyTorch Lightning for healthcare is straightforward: each tool was designed for general-purpose workloads, and adapting them to heterogeneous clinical data—EHR events, DICOM images, clinical notes, EEG waveforms—under a unified schema with consistent patient-level indexing and lazy memory semantics required substantial domain-specific engineering. Crucially, none of this backend complexity is exposed to the user, making it easy for users to extend the framework to their own workflows without needing to deeply understand how to use Dask, Polars, or PyTorch Lightning's data module directly.
>
> ---
>
> * **[Benchmarking other frameworks]** SQL-based pipelines are prohibitively heavy-handed for most academic researchers: they require data in a structured relational format and are practically incompatible with signals and imaging data. More broadly, few mature open-source healthcare AI frameworks support the same breadth of modalities, making direct comparisons difficult. If the reviewer has specific frameworks in mind, we would be happy to include them in a future revision.
>
> ---
>
> * **[Significance tests]** We will add standard deviation bars in a later revision. We hope the consumer hardware benchmarks above concretely demonstrate that PyHealth 2.0 can process large clinical datasets faster and with greater accessibility.
>
> ---
>
> * **["7 lines of code" claim]** The underlying point is technically meaningful: PyHealth 2.0 reduces every modality's data pipeline—EHR, images, signals, or text—to the same 7-step initialization structure. Research repositories commonly contain hundreds of lines of hard-to-navigate preprocessing code that is rarely reused across tasks. Standardizing this across modalities is a genuine contribution to reproducibility and accessibility.

---

> > ### Author Rebuttal · Reviewer_qwws · 2026-04-01
> >
> > I appreciate the authors' detailed rebuttal and the practical utility of the PyHealth 2.0 toolkit for clinical AI. Yet, I remain concerned with anonymity, methodological novelty, and the incomplete benchmarking, and keep my score as it is.
> >
> > Here are the reasons:
> > - The manuscript has several instances of explicit self-identification that disclose the anonymity of the review process.
> > - I remain unconvinced of the underlying engineering novelty. Even software contributions should demonstrate significant engineering innovation.
> > - The empirical evaluations are not sufficient.
> > - Significance tests are promised for future revisions.
> > - In the rebuttal, the authors provided tables of results for their own models on other modalities, but did not provide a head-to-head comparison against other baselines.

---

> > > ### Author Response · Authors · 2026-04-01
> > >
> > > While we're sorry that we were unable to change the reviewer's mind, we would like to address the specific points raised in their acknowledgement:
> > >
> > > **The manuscript has several instances of explicit self-identification that disclose the anonymity of the review process.** While PyHealth is publicly known, the paper contains no information that personally identifies the authors. We will review the manuscript and supporting details again, but we are confident that it does not violate anonymity here.
> > >
> > > **The empirical evaluations are not sufficient.** We have benchmarked across multiple hardware configurations, operating systems, datasets, and tasks, including interpretability and uncertainty quantification methods that are rarely evaluated in software papers (please see our rebuttals to Reviewer 8ZSd and Reviewer ZM1B). We are happy to provide additional benchmarks if the reviewer can provide more specific guidance on what they would like to see.
> > >
> > > **Underlying engineering novelty.** PyHealth 2.0 provides a unique contribution in the breadth of modalities and tasks supported and the flexibility of its design, especially compared to existing clinical deep learning toolkits (see Table 1). The engineering novelty is in the design and implementation of a unified, streaming-capable pipeline spanning EHR, imaging, text, and biosignals with modular baselines on a single framework that covers the entire lifecycle of a clinical deep learning project (see Figures 5–6 for explicit examples of multimodal dataloading, interpretability, and uncertainty quantification). Our laptop benchmarks demonstrate that smaller machines can process large EHR datasets effectively, an aspect of accessibility that has not been seriously considered before and that broadens clinical deep learning to those without large compute systems. To our knowledge, no other framework provides this breadth of support across modalities and tasks with this level of efficiency, particularly in a space where existing software is either narrow in scope or ignores the full clinical modeling lifecycle.
> > >
> > > **Significance tests.** Our benchmarks across multiple hardware devices and operating systems provide a meaningful picture of result variability. A controlled head-to-head comparison against other frameworks is not feasible for ETL performance because competing tools fail with out-of-memory errors on the same laptop hardware. While we will rerun benchmarks on the larger workstation in a future revision, significance tests are not necessary to establish PyHealth 2.0's ETL performance as the OOM failures of competing frameworks are themselves the result (and the absolute magnitude of the difference in performance here makes significance tests less meaningful). For model performance benchmarks, we will add significance tests in a future revision, but we do provide standard deviations across our conformal prediction metrics. Our original goal was to show that it was possible to quickly benchmark models across multiple tasks and modalities with PyHealth 2.0.
> > >
> > > **Head-to-head comparison against other baselines.** We benchmarked PyHealth 2.0's ETL performance against other frameworks in the original submission on the same device. The models in PyHealth (EHRMamba, Adacare, Retain, BioBERT, and others) are community contributions; PyHealth 2.0's goal is not to introduce new models but to provide a unified, extensible framework for applying and comparing them across modalities. The reviewer's request for a head-to-head baseline comparison does not specify explicitly which baselines they would like to see. We are happy to provide additional benchmarks given more specific guidance.

---

### Official Review · Reviewer_BNxN · 2026-03-12

**Soundness:** 4
**Presentation:** 4
**Significance:** 4
**Originality:** 4
**Overall Recommendation:** 5
**Confidence:** 4

**Summary:**

PyHealth 2.0 is a clinical deep learning toolkit designed to reduce reproducibility failures due to inconsistent preprocessing, computational barriers from large EHR datasets such as MIMIC-IV, and domain knowledge gaps between ML and clinical communities. The system supports EHRs, biosignals, imaging, and clinical texts within a single dataloader.

**Compliance With Llm Reviewing Policy:**

Affirmed.

**Key Questions For Authors:**

1. Can you provide an actual laptop benchmark (16GB RAM, consumer CPU) for at least one task, rather than inferring laptop feasibility from the workstation memory plot?
2. Table 6 shows patient random access at 208.85ms for PyHealth 2.0 vs. 0.01ms for PyHealth 1.16. For tasks involving iterative patient-level queries during training, how much does this affect total training time? Is the caching mechanism sufficient to make this irrelevant in practice?
3. The multimodal support is described as a key contribution, but Section 5 acknowledges that off-the-shelf multimodal models with missing modalities are not supported. How many of the 25+ models in the library work with the multimodal dataloader?
4. Conformal prediction under distribution shift is flagged as a known limitation. Given that patient distribution shift is endemic in clinical settings, how should users interpret uncertainty estimates from the current implementation when deploying in a new patient population?

**Limitations:**

The paper should explicitly state that PyHealth does not implement any data governance or de-identification tooling; therefore, users working with real patient data must manage regulatory compliance such as HIPAA and GDPR independently.

**Strengths And Weaknesses:**

Strengths:
1. Extensive benchmarks with different memory profiles.
2. The five-step pipeline framing at the start of the introduction provides readers with an immediate anchor.
3.The lazy-loading architecture and unified multimodal dataloader are technically novel elements.
4. The toolkit has a significant and practical value in clinical AI community.

Weaknesses:
1. The benchmark hardware (AMD EPYC 7513, 1TB RAM) is a high-end workstation. The claim that the system fits within a 16GB laptop is supported by Figure 3 showing flat memory usage; however, no actual laptop benchmark is reported.
2. The code reduction claims require a precise definition. Table 2 notes that savings for ML tasks stem from pre-implemented logic, meaning the comparison may not be fair against frameworks that also pre-implement those tasks.
3. The multimodal model support is self-described as highly limited in Section 5; however, the abstract and introduction feature multimodal support as a headline contribution.
4. The conditions under which each system achieves scalability should be stated uniformly in Table 1.

---

> ### Author Rebuttal · Authors · 2026-03-29
>
> * **[Laptop benchmark]** Yes, we benchmarked the full `labevents` mortality task on a MacBook Pro M2 Pro (16GB) and ASUS Zephyrus G16 (32GB DDR5). Both outperform our workstation due to: (1) NAS-based I/O bottlenecks on the workstation vs. local NVMe SSDs on laptops; (2) higher memory bandwidth on both laptops vs. DDR4; and (3) higher single-core performance, which is the relevant metric for our sequential streaming workload. We will revise our manuscript to contain these results and note in the limitations that performance may vary across hardware configurations.
>
> **MIMIC-IV `labevents` Mortality — Data Processing Benchmarks (`num_workers = 8`)**
>
> | Device | Wall Time (s) | Peak Mem (GB) |
> | :----- | ----: | ----: |
> | Large Workstation (NAS, DDR4) | 2,975 | 23.57 |
> | MacBook Pro M2 Pro (16GB) | 1,357 | 7.43 |
> | ASUS Zephyrus G16 (32GB DDR5) | 1,604.65 | 19.67 |
> | Pandas (baseline) | 93,708 | 49.23 |
>
> **(`num_workers = 4`)**
>
> | Device | Wall Time (s) | Peak Mem (GB) |
> | :----- | ----: | ----: |
> | Large Workstation (NAS, DDR4) | 4,605 | 23.39 |
> | MacBook Pro M2 Pro (16GB) | 2,009.53 | 5.31 |
> | ASUS Zephyrus G16 (32GB DDR5) | 2,617.41 | 19.15 |
> | Pandas (baseline) | 93,708 | 49.23 |
>
> ---
>
> * **[Code reduction claim fairness]** Fair criticism—we will revise Table 2. Below is a direct line count comparison per task. PyHealth 2.0 is slightly higher than 1.16 because (1) tasks explicitly define input/output schemas for readability, and (2) tasks are OOP rather than functional (as in Pandas, MEDS, and 1.16), giving each variant a unique identifier, which is crucial for reproducibility. The count remains comparable to 1.16, and crucially every pipeline variant shares the same 7-line initialization structure—we expect this standardization to pay off in maintainability and extensibility as the number and types of tasks and modalities grow.
>
> | Total Lines | Patient Exploration | Mortality (Labevents) | Length of Stay | Drug Rec. |
> | :---- | :---- | :---- | :---- | :---- |
> | Pandas | 16 | 51 | 22 | 24 |
> | PyHealth 1.16 | 14 | **27** | **14** | **16** |
> | MEDS | 12 | 43 | 38 | 39 |
> | PyHealth 2.0 | **10** | 34 | 18 | 23 |
>
> ---
>
> * **[Scalability conditions in Table 1]** The key differentiator is sharding and dynamic distribution across disk and memory. All frameworks outside MEDS and PyHealth 2.0 assume full in-memory loading, causing failures on memory-constrained systems. We will revise the wording to clarify that scalability applies to consumer hardware with limited memory, and that all frameworks can scale on workstations with sufficient RAM.
>
> ---
>
> * **[Multimodal support scope]** We will clarify that PyHealth 2.0 supports multimodal *dataloading* but not yet multimodal models—all current models are specialist (EHR, vision, or signal). Multimodal models are in active development for a future release.
>
> ---
>
> * **[Patient random access latency and its effects on task processing]** The 208.85ms latency applies only to ad-hoc random access during an exploration phase. During task processing, patients are iterated sequentially to construct sample tensors, so this latency is not incurred. After processing, samples are cached on disk, making subsequent training reads fast with no patient-level random access.
>
> ---
>
> * **[Conformal prediction under distribution shift]** This is an open problem with limited prior work on large healthcare datasets. From our preliminary evaluations, coverage is heavily dependent on data splits. When patients overlap between training and inference, personalized conformal prediction methods work well and reduce coverage gaps. When the inference or test population is fully disjoint from training, coverage becomes much harder to maintain—it is not clear that existing shift adjustments or invariance-based methods provide meaningful improvement in that regime. Fortunately, PyHealth implements coverage metrics directly, so users can empirically test whether conformal prediction is providing useful coverage in their deployment setting. As a reference, our current TUEV results below are under random splits (where some patient overlap exists):
>
> **Conformal Prediction on TUEV (α=0.1, random sample splits)**
>
> | Method | Empirical Coverage | Set Size |
> | :---- | :---- | :---- |
> | Conformal Prediction | 0.7461 ± 0.0334 | 1.11 ± 0.21 |
> | + KDE Covariate Shift Adjustment | 0.7457 ± 0.0072 | 1.11 ± 0.14 |
> | + KMeans Adjustment | 0.7488 ± 0.0152 | 1.23 ± 0.24 |
> | Neighborhood Conformal Prediction | 0.9152 ± 0.0117 | 1.25 ± 0.13 |
>
> ---
>
> * **[Limitations: data governance]** We will explicitly state in the limitations section that PyHealth does not implement data governance or de-identification tooling, and that users must manage regulatory compliance (HIPAA, GDPR, etc.) independently.

---

> > ### Author Rebuttal · Reviewer_BNxN · 2026-04-01
> >
> > The authors have adequately addressed all the key questions and concerns I had regarding the paper. The revised paper should reflect all the responses and include all the rebuttal details provided by the authors.

---

> > > ### Author Response · Authors · 2026-04-01
> > >
> > > Thank you very much! Thank you for your service in reviewing our paper!

---

### Official Review · Reviewer_8ZSd · 2026-03-12

**Soundness:** 3
**Presentation:** 3
**Significance:** 2
**Originality:** 1
**Overall Recommendation:** 3
**Confidence:** 2

**Summary:**

The authors propose PyHealth 2.0, a toolkit that provides a unified framework that integrates multiple modalities and datasets, different models, and evaluation metrics.

It is an extension of PyHealth 1.16, compared to which the new version supports multimodal data integration, provides an expanded model library and post-hoc deployment tools for interpretability and uncertainty quantification, and "_Most significantly, PyHealth 2.0 addresses the memory management issues identified by Steinberg et al. (2024) in PyHealth 1.16, enabling users to train clinical predictive models on consumer-grade hardware where memory is limited_".

**Compliance With Llm Reviewing Policy:**

Affirmed.

**Final Justification:**

I decreased my confidence from 3 to 2 to reflect the fact that I wasn't aware there's a specific track for software libraries and that this is not my main area of expertise.

Personally, I would however expect a paper that proposes a new library to still apply this library in order to extract more clear research-relevant findings, not solely describe the underlying engineering work. Therefore, I maintain my score of "3: Weak reject".

**Key Questions For Authors:**

Minor things:
- Figure 2 caption, "For post-training utility, we provide g. model interpretation tools" --> "For post-training utility, we provide (g) model interpretation tools"?
- Figure 4 should probably be a pdf version instead.

**Limitations:**

Yes.

**Strengths And Weaknesses:**

Strengths:
- The paper is well written, contains basically not a single typo or similar issue.
- PyHealth 2.0 seems like a good update of PyHealth 1.16, and I'm sure this toolkit could be useful and help researchers in practice.






Weaknesses:
- PyHealth 2.0 is just an update to a preexisting toolkit, mainly expanding its functionality in terms of additional types of datasets, models, and post-hoc deployment tools. While this definitely can be useful, the technical/methodological research contribution itself seems limited.

- The only actual results presented in the main paper are Figure 3 and Table 2. The toolkit is not used to e.g. benchmark and compare common models on some clinically relevant task.

- I'm sure PyHealth can be a useful tool, and I definitely want to encourage this type of work, but I'm just not entirely sure that an ICML research paper is the appropriate venue/format for this. I don't know what I learned by reading this paper, other than that I probably should explore whether PyHealth could be useful in my own work. I don't quite see how the paper itself provides any clear interesting findings/insights or actionable takeaways.

- Is this even an anonymous submission? E.g., _"To bridge this gap, we introduced RHealth"_.

---

> ### Author Rebuttal · Authors · 2026-03-29
>
> * **[Limited contribution / venue fit]** PyHealth 2.0 is a relevant/valid submission under the software library track that we have submitted to, (see https://icml.cc/Conferences/2026/CallForPapers).
>
> Nonetheless, PyHealth 2.0's contribution is engineering: it is the only open-source framework unifying EHR, image, text, and biosignal processing under a single Dask-based streaming pipeline that runs efficiently on consumer hardware to the best of our knowledge. The benchmarks below illustrate the practical utility of this—not only can researchers scale data processing across all clinical data types, but they can also benchmark post-hoc interpretability methods and uncertainty quantification frameworks such as conformal prediction within the same pipeline, tasks that were previously fragmented across different repositories and are now accessible with a few lines of PyHealth code. As an open-source framework lowering the barrier to healthcare ML research, we believe PyHealth 2.0 is a strong fit for the ICML software track.
>
> ---
>
> * **[No benchmark results in the paper]** We apologize for omitting these results; below we provide benchmarks across EHR, image, text, and biosignal tasks, showing that default settings yield reasonable, reproducible baselines.¹ Full data processing benchmarks across laptops and workstations are in our responses to Reviewers qwws and ZM1B. Additional datasets (MIMIC-III, eICU, etc.) are omitted for space but will be included in a later revision.
>
> **EHR — MIMIC-IV Mortality**²
>
> | Model | AUROC | PRAUC |
> | :---- | :---- | :---- |
> | RNN | 0.680 | 0.047 |
> | Retain | 0.610 | 0.038 |
> | Adacare | 0.680 | 0.044 |
>
> **EHR — MIMIC-IV Length of Stay**
>
> | Model | Acc | F1-Ma | Loss |
> | :---- | :---- | :---- | :---- |
> | RNN | 0.457 | 0.405 | 1.375 |
> | Retain | 0.434 | 0.379 | 1.436 |
> | Transformer | 0.420 | 0.353 | 1.549 |
> | Deepr | 0.414 | 0.351 | 1.495 |
> | EHRMamba | 0.400 | 0.335 | 1.548 |
> | Adacare | 0.383 | 0.314 | 1.604 |
>
> **EHR — MIMIC-IV Drug Recommendation**
>
> | Model | Jaccard | F1-Sa | Loss |
> | :---- | :---- | :---- | :---- |
> | EHRMamba | 0.492 | 0.650 | 0.060 |
> | Deepr | 0.481 | 0.641 | 0.062 |
> | Transformer | 0.466 | 0.627 | 0.063 |
> | Adacare | 0.466 | 0.627 | 0.063 |
> | RNN | 0.430 | 0.593 | 0.068 |
> | GameNet | 0.428 | 0.591 | 0.069 |
> | Retain | 0.423 | 0.587 | 0.070 |
>
> **Image — Covid19CXR Classification**
>
> | Model | Acc | F1-Ma | AUC |
> | :---- | :---- | :---- | :---- |
> | CNN | 0.865 | 0.874 | 0.975 |
> | ResNet-18 | 0.953 | 0.960 | 0.994 |
> | ViT-B/32 | 0.888 | 0.893 | 0.977 |
>
> **Text — Medical Transcriptions Classification**
>
> | Model | Acc | F1-Ma |
> | :---- | :---- | :---- |
> | Embed | 0.340 | 0.066 |
> | BERT Base | 0.362 | 0.047 |
> | BioBERT | 0.370 | 0.072 |
>
> **Signals — TUAB Abnormal EEG Detection**
>
> | Model | AUC | Acc | F1-Ma |
> | :---- | :---- | :---- | :---- |
> | BIOT | 0.892 | 0.810 | 0.770 |
> | ContraWR | 0.833 | 0.762 | 0.728 |
> | SPaRCNet | 0.855 | 0.768 | 0.700 |
> | TFM-Tokenizer | 0.890 | 0.811 | 0.778 |
>
> **Signals — TUEV EEG Event Classification**
>
> | Model | Acc | F1-Ma |
> | :---- | :---- | :---- |
> | BIOT | 0.716 | 0.422 |
> | ContraWR | 0.741 | 0.503 |
> | SPaRCNet | 0.691 | 0.364 |
> | TFM-Tokenizer | 0.777 | 0.545 |
>
> **Interpretability — Mortality (Transformer)**
>
> | Method | Comp. | Suff. |
> | :---- | :---- | :---- |
> | Integrated Gradient | 0.601 | 0.029 |
> | DeepLIFT | 0.323 | 0.157 |
> | GIM | 0.571 | 0.039 |
> | SHAP | 0.480 | 0.116 |
> | LIME | 0.522 | 0.148 |
> | Attn-Grad | **0.603** | **-0.008** |
>
> **UQ — Conformal Prediction on TUEV (α=0.1, random splits)**
>
> | Method | Coverage | Set Size |
> | :---- | :---- | :---- |
> | CP | 0.746 ± 0.033 | 1.11 ± 0.21 |
> | + KDE Shift Adj. | 0.746 ± 0.007 | 1.11 ± 0.14 |
> | + KMeans Adj. | 0.749 ± 0.015 | 1.23 ± 0.24 |
> | Neighborhood CP | 0.915 ± 0.012 | 1.25 ± 0.13 |
>
> ¹ All models: 20 epochs, lr=1e-4, AdamW; no hyperparameter tuning or pretraining.
> ² Additional datasets (eICU, MIMIC-III, etc.) and models omitted for space; we will include the full set in a later revision.
>
> ---
>
> * **[Anonymity concern]** We tried our best to anonymize explicit details, noting that the full software release is publicly available. We can confirm that the authors of this submission are largely disjoint from the authors of the cited prior work.
>
> ---
>
> * **[Figure 2 caption typo]** Thanks for the catch—we will fix this.
>
> * **[Figure 4 resolution]** Agreed, we will update to a PDF in a later revision.

---

> > ### Author Rebuttal · Reviewer_8ZSd · 2026-04-04
> >
> > Thank you for the response.
> >
> > I must admit that I wasn't aware that there's a specific track for software libraries etc. If so, I will have to lower the confidence of my review because that's definitely not my main area of expertise.
> >
> > Personally, I would expect a paper that proposes a new library to still apply this library in order to extract some kind of research-relevant findings, not solely describe the underlying engineering work. I just don't know what I learned by reading this paper.
> >
> > _"[Anonymity concern] We tried our best to anonymize explicit details, noting that the full software release is publicly available"_: To me, this seems like a good argument for why ICML isn't an appropriate venue/format for this type of work then. And, while I think it could be possible to maintain formal anonymity even when a previous version of the library is publicly available, the _"To bridge this gap, we introduced RHealth"_ reference seems like an explicit breach of anonymity that I don't think would be accepted for a regular ICML submission.

---

> > > ### Author Response · Authors · 2026-04-04
> > >
> > > While we're sorry that we were unable to change the reviewer's mind, we would like to address the specific points raised in their acknowledgement:
> > >
> > > **Wanting research-relevant findings.** We felt that many of the research-relevant findings that we could extract from the library would be out of scope for a software library paper, as the library's breadth enables a wide range of different research questions. That said, we would like to highlight some insights from our benchmark results: it is substantially easier to run not only predictive modeling experiments but also interpretability and uncertainty quantification experiments across different data modalities and clinical tasks with PyHealth 2.0. From our results alone, some interesting future research directions include understanding why interpretability techniques for LLMs do not readily transfer to time-series EHR modeling, why conformal prediction is more effective for some tasks than others, and why personalization can help in cases of unidentified distribution shift. We hope that by providing a unified framework for these different tasks, we can enable researchers to explore these questions more easily.
> > >
> > > **Anonymity concern.** We apologize for this oversight. We want to note that the main authors of this work are disjoint from the authors of the original PyHealth 1.16 and RHealth works, though we acknowledge they operate under the same research community. We apologize for the misleading wording.
> > >
> > >
> > > **Research Fit.** We also note that comparable works such as ACES ([ICLR 2025](https://openreview.net/forum?id=P4XmKjXTrM)), which covers cohort extraction, data processing, and benchmarking across clinical modalities within the MEDS community, have found a home at top ML venues. We believe PyHealth 2.0 is similarly well-suited for the ICML software track given its engineering contributions and practical utility for healthcare ML research. We understand the reviewer's perspective and appreciate the discussion.

---

### Official Review · Reviewer_ZM1B · 2026-03-13

**Soundness:** 3
**Presentation:** 3
**Significance:** 4
**Originality:** 3
**Overall Recommendation:** 5
**Confidence:** 4

**Summary:**

This paper presents PyHealth 2.0, an open-source toolkit for clinical deep learning that aims to improve accessibility, reproducibility, and efficiency. The framework unifies a broad range of clinical datasets, tasks, models, modalities, interpretability tools, and uncertainty methods under a single interface. The paper also reports substantial improvements in processing speed and memory usage, making large-scale clinical ML workflows more practical.

**Compliance With Llm Reviewing Policy:**

Affirmed.

**Final Justification:**

I maintain my accept recommendation. This paper is a sound and practically significant software contribution to clinical ML, with clear value in its unified framework and meaningful efficiency gains.

My main concerns were limited algorithmic novelty, stronger evidence for reproducibility, fairness of efficiency comparisons, and clarity on component maturity. The rebuttal addressed these points satisfactorily and increased my confidence in the paper, though it did not materially change my evaluation.

Overall, the originality is mainly in integration and engineering, but the practical significance is high, and the rebuttal reinforced my positive assessment.

**Key Questions For Authors:**

- Can the authors provide more direct evidence that PyHealth 2.0 improves reproducibility in practice, beyond unifying pipelines?
- Were all efficiency comparisons conducted under equally optimized settings across frameworks?
- Can the authors better clarify which components of the toolkit are the most mature and fully validated?

**Limitations:**

yes

**Strengths And Weaknesses:**

Strengths
- Addresses an important and practical problem in clinical ML: reproducibility and accessibility.
- Provides a broad and unified framework covering datasets, tasks, models, and multimodal data.
- Shows clear practical value, especially in reducing engineering burden and hardware requirements.
- Efficiency gains in runtime and memory are meaningful and well aligned with the paper’s goals.
- Likely to be useful for the community as a shared infrastructure resource.

Weaknesses
- The main contribution is largely engineering/integration, with limited algorithmic novelty.
- The evaluation focuses more on efficiency and coverage than on downstream scientific impact.
- Some broader claims about reproducibility and accessibility would benefit from more direct quantitative evidence.
- The paper is generally clear, but the breadth of coverage makes the key technical contributions slightly less sharp.

Overall assessment
- Soundness: solid for a systems/toolkit paper; the main claims are mostly supported.
- Presentation: clear overall, though somewhat feature-heavy.
- Significance: high practical significance for clinical ML researchers.
- Originality: moderate; novelty mainly comes from integration, scale, and engineering.

---

> ### Author Rebuttal · Authors · 2026-03-29
>
> * **[Limited algorithmic novelty]** We agree; this contribution fits squarely within ICML's software track, where engineering contributions are a primary focus. We hope PyHealth lowers the barrier for future algorithmic work across a range of clinically relevant tasks and modalities.
>
> ---
>
> * **[Downstream benchmarks and reproducibility evidence]** Below we provide benchmarks across EHR, image, text, and biosignal modalities as direct evidence of PyHealth 2.0's utility as a reproducible benchmarking tool.¹ Standardized task schemas and fixed default configurations make it straightforward to replicate many baselines, including various interpetability approaches and uncertainty quantification methods.
>
> **EHR (MIMIC-IV)**
>
> *Mortality*
>
> | Model | AUROC | PRAUC |
> | :---- | :---- | :---- |
> | RNN | 0.680 | 0.047 |
> | Retain | 0.610 | 0.038 |
> | Adacare | 0.680 | 0.044 |
>
> *Length of Stay*
>
> | Model | Acc | F1-Ma | Loss |
> | :---- | :---- | :---- | :---- |
> | RNN | 0.457 | 0.405 | 1.375 |
> | Retain | 0.434 | 0.379 | 1.436 |
> | Transformer | 0.420 | 0.353 | 1.549 |
> | Deepr | 0.414 | 0.351 | 1.495 |
> | EHRMamba | 0.400 | 0.335 | 1.548 |
> | Adacare | 0.383 | 0.314 | 1.604 |
>
> *Drug Recommendation*
>
> | Model | Jaccard | F1-Sa | Loss |
> | :---- | :---- | :---- | :---- |
> | EHRMamba | 0.492 | 0.650 | 0.060 |
> | Deepr | 0.481 | 0.641 | 0.062 |
> | Transformer | 0.466 | 0.627 | 0.063 |
> | Adacare | 0.466 | 0.627 | 0.063 |
> | RNN | 0.430 | 0.593 | 0.068 |
> | GameNet | 0.428 | 0.591 | 0.069 |
> | Retain | 0.423 | 0.587 | 0.070 |
>
> **Image — Covid19CXR**
>
> | Model | Acc | F1-Ma | AUC |
> | :---- | :---- | :---- | :---- |
> | CNN | 0.865 | 0.874 | 0.975 |
> | ResNet-18 | 0.953 | 0.960 | 0.994 |
> | ViT-B/32 | 0.888 | 0.893 | 0.977 |
>
> **Text — Medical Transcriptions**
>
> | Model | Acc | F1-Ma |
> | :---- | :---- | :---- |
> | Embed | 0.340 | 0.066 |
> | BERT Base | 0.362 | 0.047 |
> | BioBERT | 0.370 | 0.072 |
>
> **Signals**
>
> *TUAB — Abnormal EEG*
>
> | Model | AUC | Acc | F1-Ma |
> | :---- | :---- | :---- | :---- |
> | BIOT | 0.892 | 0.810 | 0.770 |
> | ContraWR | 0.833 | 0.762 | 0.728 |
> | SPaRCNet | 0.855 | 0.768 | 0.700 |
> | TFM-Tokenizer | 0.890 | 0.811 | 0.778 |
>
> *TUEV — EEG Events*
>
> | Model | Acc | F1-Ma |
> | :---- | :---- | :---- |
> | BIOT | 0.716 | 0.422 |
> | ContraWR | 0.741 | 0.503 |
> | SPaRCNet | 0.691 | 0.364 |
> | TFM-Tokenizer | 0.777 | 0.545 |
>
> **Interpretability — Mortality (Transformer)**
>
> | Method | Comp. | Suff. |
> | :---- | :---- | :---- |
> | Integrated Gradient | 0.601 | 0.029 |
> | DeepLIFT | 0.323 | 0.157 |
> | GIM | 0.571 | 0.039 |
> | SHAP | 0.480 | 0.116 |
> | LIME | 0.522 | 0.148 |
> | Attn-Grad | **0.603** | **-0.008** |
>
> **UQ — Conformal Prediction on TUEV (α=0.1)**
>
> | Method | Coverage | Set Size |
> | :---- | :---- | :---- |
> | CP | 0.746 ± 0.033 | 1.11 ± 0.21 |
> | + KDE Shift Adj. | 0.746 ± 0.007 | 1.11 ± 0.14 |
> | + KMeans Adj. | 0.749 ± 0.015 | 1.23 ± 0.24 |
> | Neighborhood CP | 0.915 ± 0.012 | 1.25 ± 0.13 |
>
> ¹ All models: 20 epochs, lr=1e-4, AdamW; no hyperparameter tuning or pretraining.
>
> ---
>
> * **[Core contributions and consumer hardware accessibility]** We will sharpen the focus around two core contributions: (1) a streaming data processing engine that runs efficiently on consumer hardware, and (2) a unified framework for EHR, image, text, and biosignal data processing and modelingin the clinical AI space. The benchmarks below illustrate (1) concretely: PyHealth 2.0 processes the full `labevents` task faster and with less memory than a 32-core workstation on a 16GB MacBook Pro, and the benchmarks above hopefully illustrate (2).
>
> **MIMIC-IV `labevents` Mortality — Data Processing Benchmarks (`num_workers = 8`)**
>
> | Device | Wall Time (s) | Peak Mem (GB) |
> | :----- | ----: | ----: |
> | Large Workstation (NAS, DDR4) | 2,975 | 23.57 |
> | MacBook Pro M2 Pro (16GB) | 1,357 | 7.43 |
> | ASUS Zephyrus G16 (32GB DDR5) | 1,604.65 | 19.67 |
> | Pandas (baseline) | 93,708 | 49.23 |
>
> **(`num_workers = 4`)**
>
> | Device | Wall Time (s) | Peak Mem (GB) |
> | :----- | ----: | ----: |
> | Large Workstation (NAS, DDR4) | 4,605 | 23.39 |
> | MacBook Pro M2 Pro (16GB) | 2,009.53 | 5.31 |
> | ASUS Zephyrus G16 (32GB DDR5) | 2,617.41 | 19.15 |
> | Pandas (baseline) | 93,708 | 49.23 |
>
> ---
>
> * **[Efficiency comparisons normalized across frameworks]** We normalized all comparisons by worker count and used the recommended default settings from each framework's authors. All benchmarks in the processing figure were run on the same hardware to ensure a fair comparison.
>
> ---
>
> * **[Maturity of toolkit components]** Data/task processing and evaluation tooling are our most mature components. For modeling: EHR is fully validated, EEG/waveform well-supported, and clinical imaging, text, and genomics have limited support. The interpretability module is currently EHR- and imaging-only. We have an active working group expanding both, and will document these limitations in the paper.

---

> > ### Author Rebuttal · Reviewer_ZM1B · 2026-04-05
> >
> > Thanks for your reply, I mantain my positive score.

---

### Decision · Program_Chairs · 2026-04-30

**Decision:**

Accept (regular)

**Comment:**

This paper describes PyHealth 2.0, a novel software package for clinical deep learning. Reviewers praised its comprehensiveness and usability, noting that it provides a standardized, open-source pipeline for clinical deep learning that supports diverse datasets, including MIMIC-III and OMOP. Reviewers also highlighted its modularity, reproducibility, and extensive documentation as major strengths that lower the entry barrier for healthcare researchers. Persistent concerns, even after the rebuttal, include the paper's limited methodological novelty, acceptable as a software infrastructure paper; limited innovation at the software and system levels; limited or unfair evaluations; and potential violations of anonymity. It is clear that the reviewers have different interpretations and expectations of a software infrastructure paper, and their final opinions do not converge. Overall, the current AC considers that PyHealth 2.0’s impact and strengths outweigh the limitations of the current paper, and it is a significant contribution to the ML for health community, successfully bridging the gap between raw medical data and deep learning models.